# ARS2/MAGL signaling in glioblastoma stem cells promotes self-renewal and M2-like polarization of tumor-associated macrophages

Jinlong Yin [1,2,3,13✉], Sung Soo Kim [2,3,13], Eunji Choi[4,13], Young Taek Oh [3,5,6,13], Weiwei Lin[3], Tae-Hoon Kim[3], Jason K. Sa [5,7], Jun Hee Hong[3], Se Hwan Park [2], Hyung Joon Kwon[4], Xiong Jin[8,12], Yeonhee You[2], Ji Hye Kim[9], Hyunggee Kim[8], Jaekyoung Son[9], Jeongwu Lee[10], Do-Hyun Nam[5], Kui Son Choi [4], Bingyang Shi[1], Ho-Shin Gwak[4], Heon Yoo[2,3], Antonio Iavarone[6✉], Jong Heon Kim [2,11✉] & Jong Bae Park [2,3✉]

The interplay between glioblastoma stem cells (GSCs) and tumor-associated macrophages (TAMs) promotes progression of glioblastoma multiforme (GBM). However, the detailed molecular mechanisms underlying the relationship between these two cell types remain unclear. Here, we demonstrate that ARS2 (arsenite-resistance protein 2), a zinc finger protein that is essential for early mammalian development, plays critical roles in GSC maintenance and M2-like TAM polarization. ARS2 directly activates its novel transcriptional target *MGLL*, encoding monoacylglycerol lipase (MAGL), to regulate the self-renewal and tumorigenicity of GSCs through production of prostaglandin $E_2$ ($PGE_2$), which stimulates β-catenin activation of GSC and M2-like TAM polarization. We identify M2-like signature downregulated by which MAGL-specific inhibitor, JZL184, increased survival rate significantly in the mouse xenograft model by blocking $PGE_2$ production. Taken together, our results suggest that blocking the interplay between GSCs and TAMs by targeting ARS2/MAGL signaling offers a potentially novel therapeutic option for GBM patients.

[1] Henan and Macquarie University Joint Centre for Biomedical Innovation, School of Life Sciences, Henan University, Kaifeng, Henan, China. [2] Department of Cancer Biomedical Science, Graduate School of Cancer Science and Policy, National Cancer Center, Goyang, Republic of Korea. [3] Rare Cancer Branch, Research Institute and Hospital, National Cancer Center, Goyang, Republic of Korea. [4] Department of Cancer Control and Population Health, Graduate School of Cancer Science and Policy, National Cancer Center, Goyang, Republic of Korea. [5] Department of Health Sciences and Technology, SAIHST, Sungkyunkwan University, Seoul, Republic of Korea. [6] Institute for Cancer Genetics, Columbia University Medical Center, New York, NY, USA. [7] Department of Biomedical Sciences, Korea University College of Medicine, Seoul, Korea. [8] Department of Biotechnology, School of Life Sciences and Biotechnology, Korea University, Seoul, Republic of Korea. [9] Department of Biomedical Sciences, University of Ulsan College of Medicine, Seoul, Republic of Korea. [10] Department of Stem Cell Biology and Regenerative Medicine, Lerner Research Institute, Cleveland Clinic, Cleveland, OH, USA. [11] Division of Cancer Biology, Research Institute and Hospital, National Cancer Center, Goyang, Republic of Korea. [12] Present address: Laboratory of Stem Cells, NEXELCo., Ltd., Seoul, Republic of Korea. [13] These authors contributed equally: Jinlong Yin, Eunji Choi, Sung Soo Kim, Young Taek Oh. ✉email: jlyin@henu.edu.cn; ai2102@cumc.columbia.edu; jhkim@ncc.re.kr; jbp@ncc.re.kr

Glioblastoma multiforme (GBM), a rapidly growing and highly invasive cancer, is the most common and lethal malignant primary brain tumor, with a median survival of 14 months and 2-year survival rates <10%[1]. Despite the lethality and burden of the disease, there has been little progress in GBM therapeutics. Failure of GBM treatment is attributable in part to a small population of cells expressing stemness characteristics[2]. Emerging evidence suggests that these glioblastoma stem cells (GSCs) dynamically influence and communicate with multiple aspects of the GBM tumor microenvironment[3,4]. As one of the main regulatory components in the tumor microenvironment, tumor-associated macrophages (TAMs) are especially important in GBM, given their significant correlation with patient prognosis and glioma progression and grades[5,6]. Interestingly, it has been shown that TAMs and GSCs, which are co-enriched in hypoxic and perivascular niches and are increased after recurrent GBM-irradiation, show a close functional relationship in conferring tumorigenesis[7,8]. Several previous studies have suggested that TAMs can be functionally subtyped according to polarization status, namely M1 and M2, a further demonstrated that M2 TAMs in particular play a tumor-supportive role in GSCs[7,9]. Notably, preventing TAM polarization into the M2 subtype has been reported to block glioma progression and tumor growth[7,10]. Despite the convincing functional correlation between TAMs and GSCs, the molecular links defining the relationship between these two elements have not yet been defined.

Arenite resistance gene 2 (ARS2), first identified in an arsenite-resistant hamster cell line, contains multiple nuclear-localization signals and a zinc finger-like domain[11]. In addition to its contribution to arsenite resistance, ARS2 is noteworthy for its role in mammalian development, modulation of cell proliferation, and promotion of the accumulation of several micro RNAs (miRNAs) involved in cellular transformation and inflammation[12,13]. A recent report demonstrates that ARS2 is closely associated with maintaining the self-renewal identity of neural stem cells (NSCs), identifying ARS2 as one of the transcription factors that controls pluripotent NSCs through direct induction of the pluripotent-maintenance gene, SOX2[14]. However, the role of ARS2 has never been studied in the context of cancer, let alone in glioma generally or GSCs in particular.

Monoacylglycerol lipase (MAGL), encoded by the MGLL gene, is a lipolytic enzyme that hydrolyzes monoacylglycerols to glycerol and free fatty acids (FFAs)[15]. MAGL is most highly expressed in the brain and white adipose tissue; however, is also highly expressed in aggressive cancer cells, where it modulates cancer metabolism through the production of FFAs[15–17]. Another role of MAGL is to hydrolyze endocannabinoid 2-arachidonoylglycerol (2-AG) to arachidonic acid (AA), which can be enzymatically converted to prostaglandin $E_2$ ($PGE_2$)[18,19]. It has been shown that pharmacological blockade of MAGL with clinically available inhibitors exerts anti-inflammatory effects in the brain and neuroprotective effects in mouse models of various neuroinflammation-mediated diseases[20]. Despite convincing clinical evidence supporting the roles of MAGL, no studies have addressed the association of MAGL with the most fatal brain disease, GBM, and specifically GSCs. Furthermore, intriguing unanswered questions about potential regulators of MAGL remain at molecular and cellular levels.

In this study, we provide the first demonstration that ARS2 regulates the stem cell-like characteristics of GSCs through direct transcriptional activation of MAGL. ARS2-MAGL signaling activates self-renewal by inducing the accumulation of β-catenin, and exerts tumorigenic activity in mouse xenograft models of GSCs by inducing M2-like TAM polarization, both of which are mediated by MAGL-dependent production of $PGE_2$. Collectively, our findings establish MAGL as a prognostic factor in GBM, and show that pharmacological inhibition of MAGL offers potential benefit in the treatment of GBM.

## Results

**ARS2 is correlated with poor survival and GSC stemness.** To study the relationship between ARS2 and clinical outcome in glioma patients, we first analyzed the expression profile of ARS2 in the REMBRANDT (REpository for Molecular BRAin Neoplasia DaTa) database, which included data from 105 patients with astrocytoma, 181 with GBM, and 336 with all forms of glioma. ARS2 mRNA expression was significantly upregulated in glioma patients compared with that in non-tumor brain tissue from 28 patients (Fig. 1a). Among 336 patients in the all-glioma group, patients with higher expression of ARS2 exhibited significantly shorter survival than those with low expression (Fig. 1b). Notably, a similar significant relationship was also observed in 181 patients with GBM (Fig. 1c). Consistent with this, increased expression of ARS2 predicted poor prognosis among all glioma and GBM patients in the TCGA (The Cancer Genome Atlas) database (Fig. 1d, e). These results collectively reveal an important association between ARS2 mRNA expression and high-grade glioma as well as poor patient survival.

To interrogate the protein levels of ARS2 in glioma patients, we analyzed tumor tissues from 49 glioma patients and five normal brain controls from the National Cancer Center (NCC), Republic of Korea. ARS2 protein was barely detectable in normal brains, but was widely and strongly expressed in patient tumor samples (Fig. 1f and Supplementary Fig. 1a, b). Next, we examined whether ARS2 expression was relevant for stem-like properties in glioma. Immunofluorescence analyses revealed that ARS2 was coexpressed with Nestin (a marker of NSCs) in a human GSC X01-derived mouse xenograft sample (Fig. 1g, h). Moreover, significant positive correlations between ARS2 and Nestin expression in the TCGA database are plotted in Fig. 1i. Collectively, these results indicate that upregulation of ARS2 mRNA and protein is strongly associated with glioma malignancy and GSC self-renewal.

**ARS2 regulates the self-renewal and tumorigenicity of GSCs.** To determine whether ARS2 is involved in the regulation of glioma stemness, we selectively knocked down ARS2 expression in GSCs using two different short hairpin (interfering) RNAs (shRNAs), and assayed for sphere-forming ability and cell proliferation (Fig. 2a–f). Specific knockdown of ARS2 suppressed expression of Nestin, a marker of undifferentiated cells, with a concurrent increase in the expression of glial fibrillary acidic protein (GFAP), a marker of differentiation, in GSC 528 and X01 cells (Fig. 2a, d and Supplementary Fig. 2a, b). Knockdown of ARS2 significantly decreased the sphere-forming ability of GSCs in limiting dilution assays[21], a widely used method for determining the self-renewal capacity of stem cells (Fig. 2b, e). Knockdown of ARS2 also significantly blocked proliferation of GSCs (Fig. 2c, f). Conversely, overexpression of ARS2 significantly increased sphere-forming ability and proliferation rate in GSCs (578 and 0502 cells; Supplementary Fig. 2c–h).

To address the tumorigenicity of ARS2 in vivo, we created mouse xenograft models from the X01 control cell line and its ARS2-knockdown derivatives, and compared tumor mass and overall survival. Hematoxylin and eosin (H&E) staining of brain slices from tumor-bearing mice revealed a clear decrease in tumor mass in ARS2-knockdown xenograft models compared with the corresponding parental control models (Fig. 2g). Moreover, overall survival was significantly longer in both ARS2-knockdown xenografts than in control mice (Fig. 2h). Collectively, these

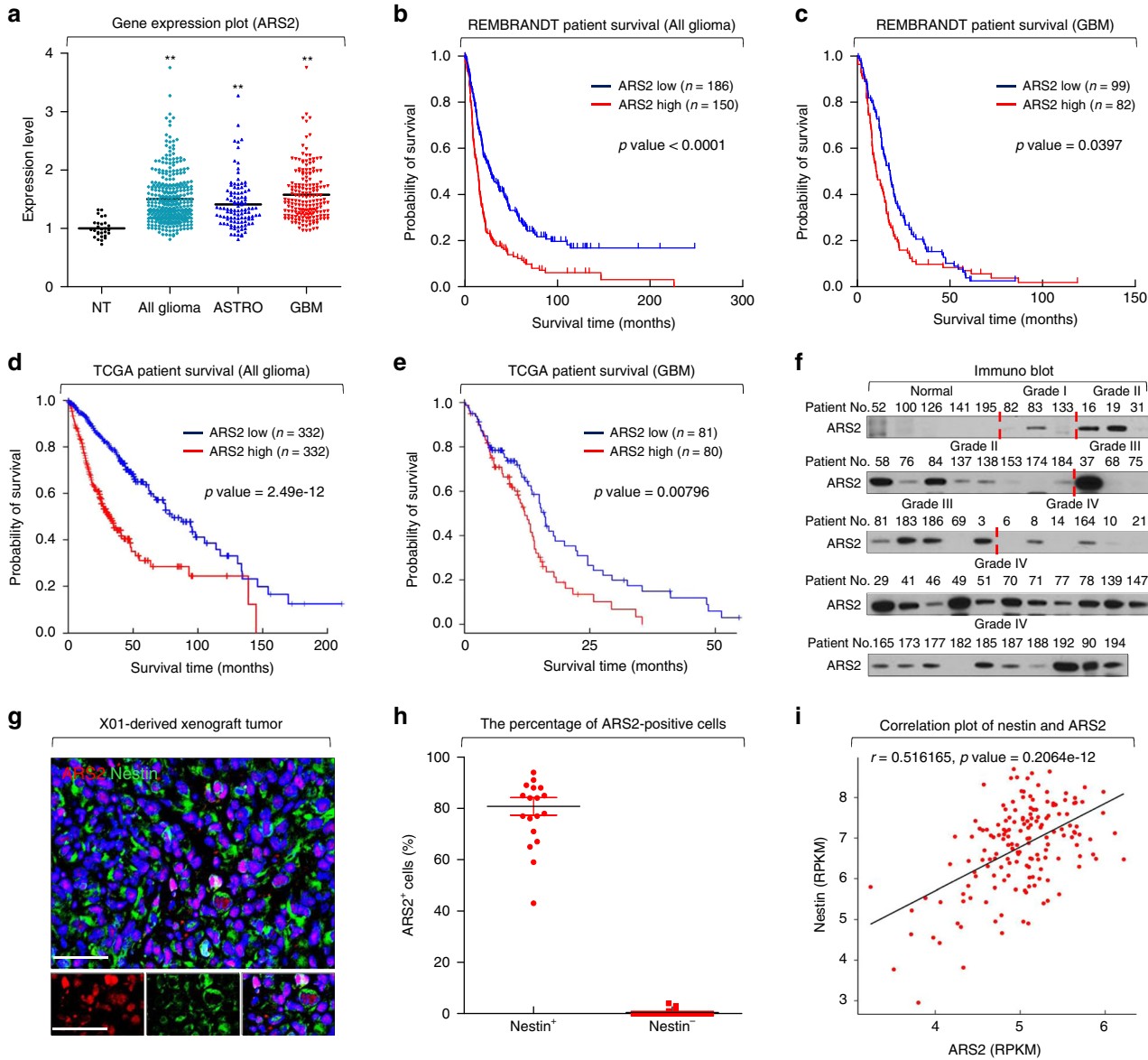

**Fig. 1 ARS2 is highly expressed in high-grade brain tumors. a** ARS2 expression in each type of brain tumor from the REMBRANDT database. **b**, **c** Kaplan–Meier survival plots for all glioma patients and GBM patients with high and low ARS2 expression. Data were obtained from the REMBRANDT of the National Cancer Institute (log-rank test). **d**, **e** Kaplan–Meier survival plots for all glioma patients and GBM patients with high (top 50% contribution) and low (down 50% contribution) ARS2 expression. Data were obtained from the TCGA database. **f** Immunoblot (IB) analysis of ARS2 in patient tissues from the National Cancer Center, Republic of Korea. GAPDH was used as a loading control[45]. **g** Representative immunofluorescence (IF) image of ARS2 and Nestin expression in GBM xenografts derived from X01 cells. Nuclei were counterstained with DAPI (blue). Scale bar, 50 µm. **h** Percentage of ARS2-positive cells among Nestin-positive and -negative cells. Lines show means and ±SD. **i** Correlation dot-plot of ARS2 and Nestin from the TCGA database ($n = 162$).

findings indicate that ARS2 is required for GSC proliferation and is essential for regulating GSC self-renewal and glioma tumorigenicity.

**Identification of *MGLL* as a novel target gene of ARS2.** Considering that ARS2 is a well-known transcriptional regulator involved in the maintenance of NSC stemness, we performed transcriptome profiling using RNA sequencing (RNA-Seq) analysis after deletion of ARS2. Each gene identified as being downregulated upon ARS2-knockdown was carefully examined for its significance in cancer pathogenesis. Genes involved in housekeeping activities or those with an inconsequential relationship with cancer were excluded. The most promising gene

downregulated upon ARS2-knockdown was *MGLL*, encoding MAGL (Fig. 3a). Although there are numerous reports of a significant association of MAGL with aggressive cancers[17,22,23], there are no studies linking MAGL with glioma.

To establish an association between MAGL and ARS2 expression, we first confirmed MAGL mRNA and protein expression in ARS2-overexpressing and -knockdown GSCs. Overexpression of ARS2 in GSCs upregulated MAGL at both mRNA and protein levels (Fig. 3b, c). Conversely, reducing the expression of ARS2 in GSCs downregulated MAGL expression at both mRNA and protein levels (Fig. 3d, e).

Using chromatin immunoprecipitation (ChIP) assays, we further examined whether *MGLL* is a direct downstream target of ARS2. To this end, we designed four primer pairs (regions 1–4)

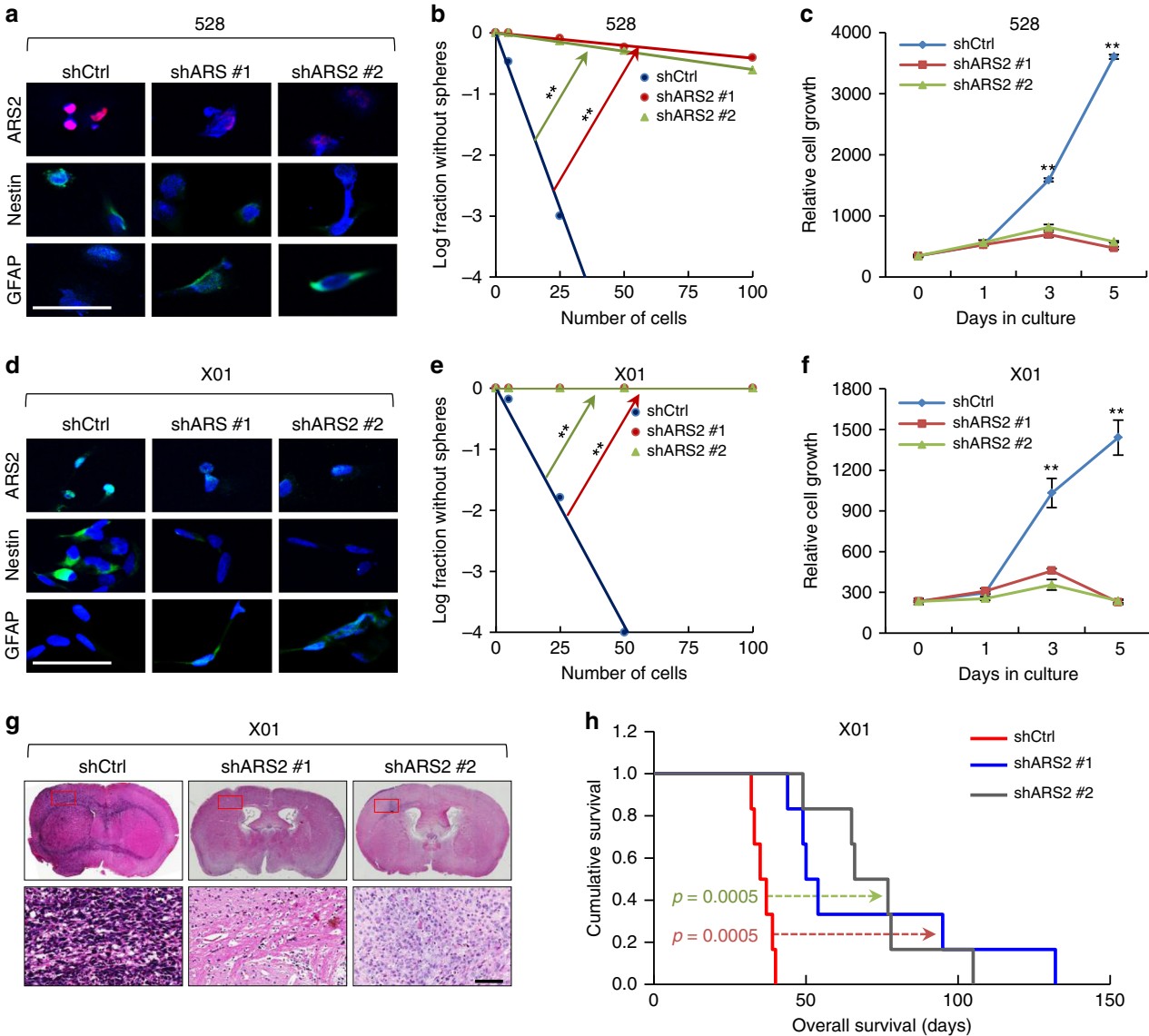

**Fig. 2 ARS2 regulates the stemness and tumorigenesis of GSCs. a** Representative immunocytochemistry (ICC) images of ARS2, Nestin, and GFAP in control 528 cells (528-shCtrl), and two different ARS2-knockdown 528 cell lines (528-shARS2 #1 and 528-shARS2 #2). Nuclei were counterstained with DAPI (blue). Scale bar, 50 μm. **b** Limiting dilution assays (LDAs) performed using 528-shCtrl, 528-shARS2 #1, and 528-shARS2 #2 cells. ($n = 24$, **$P <$ 0.01, $t$-test. **c** Cell proliferation assays performed using 528-shCtrl, 528-shARS2 #1, and 528-shARS2 #2 cells. All error bars represent mean ± standard error of the mean (SEM; $n = 3$). **$P < 0.01$, $t$-test. **d** Representative ICC images of ARS2, Nestin, and GFAP in X01-shCtrl, X01-shARS2 #1, and X01-shARS2 #2 cells. Nuclei were counterstained with DAPI (blue). Scale bar, 50 μm. **e** LDAs performed using X0-shCtrl, X01-shARS2 #1, and X01-shARS2 #2 cells. ($n = 24$, **$P < 0.01$, $t$-test). **f** Cell proliferation assays performed using X01-shCtrl, X01-shARS2 #1, and X01-shARS2 #2 cells. All error bars represent mean ± SEM ($n = 3$). **$P < 0.01$, $t$-test. **g** H&E staining of the whole brains from mice implanted with X01-shCtrl, X01-shARS2 #1, or X01-shARS2 #2 cells. Scale bar, 100 μm. The sample is extracted at 42 days after cell injection. **h** Kaplan–Meier survival plots for the orthotopic xenograft mouse model ($n = 6$; $1 \times 10^5$ cells injected per mouse, log-rank test). Median survival of the orthotopic mice injected with X01-shCtrl, X01-shARS2 #1, or X01-shARS2 #2 was 36 days, 52 days, and 71.5 days, respectively.

covering the −1300 to +26 bp region relative to the transcription start site (TSS) of *MGLL* (Supplementary Fig. 3a). As shown in supplementary Fig. 3b, antibodies against ARS2 effectively immunoprecipitated a specific region upstream of the *MGLL* gene corresponding to regions 3 (−1018 to −887 bp) and 4 (−1300 to −1093 bp). The relative enrichment of ARS2 in regions 3 and 4 was assessed by quantitative polymerase chain reaction (qPCR), which revealed 5- and 25-fold higher levels of ARS2 occupancy in regions amplified by primer sets 3 and 4, respectively, compared with immunoprecipitations with control IgG (Fig. 3f, g). To confirm the transcriptional relationship between ARS2 and *MGLL*, we used a lentiviral-based reporter system to monitor the expression of a luciferase reporter gene linked to the upstream promoter region, identified above, containing binding site(s) for ARS2 (Fig. 3h). These reporter assays were performed in GSC X01 cells, with or without co-infection with an ARS2-specific shRNA-expressing lentivirus. As shown in Fig. 3h, ARS2 knockdown in X01 cells significantly decreased relative luciferase expression, indicating reduced transcriptional activity of ARS2 toward *MGLL*. Collectively, these data demonstrate that a number of genes are potentially regulated at the transcriptional level by ARS2, and specifically identify *MGLL* as a novel target of ARS2 in GSCs, consistent with the role of MAGL in aggressive cancers.

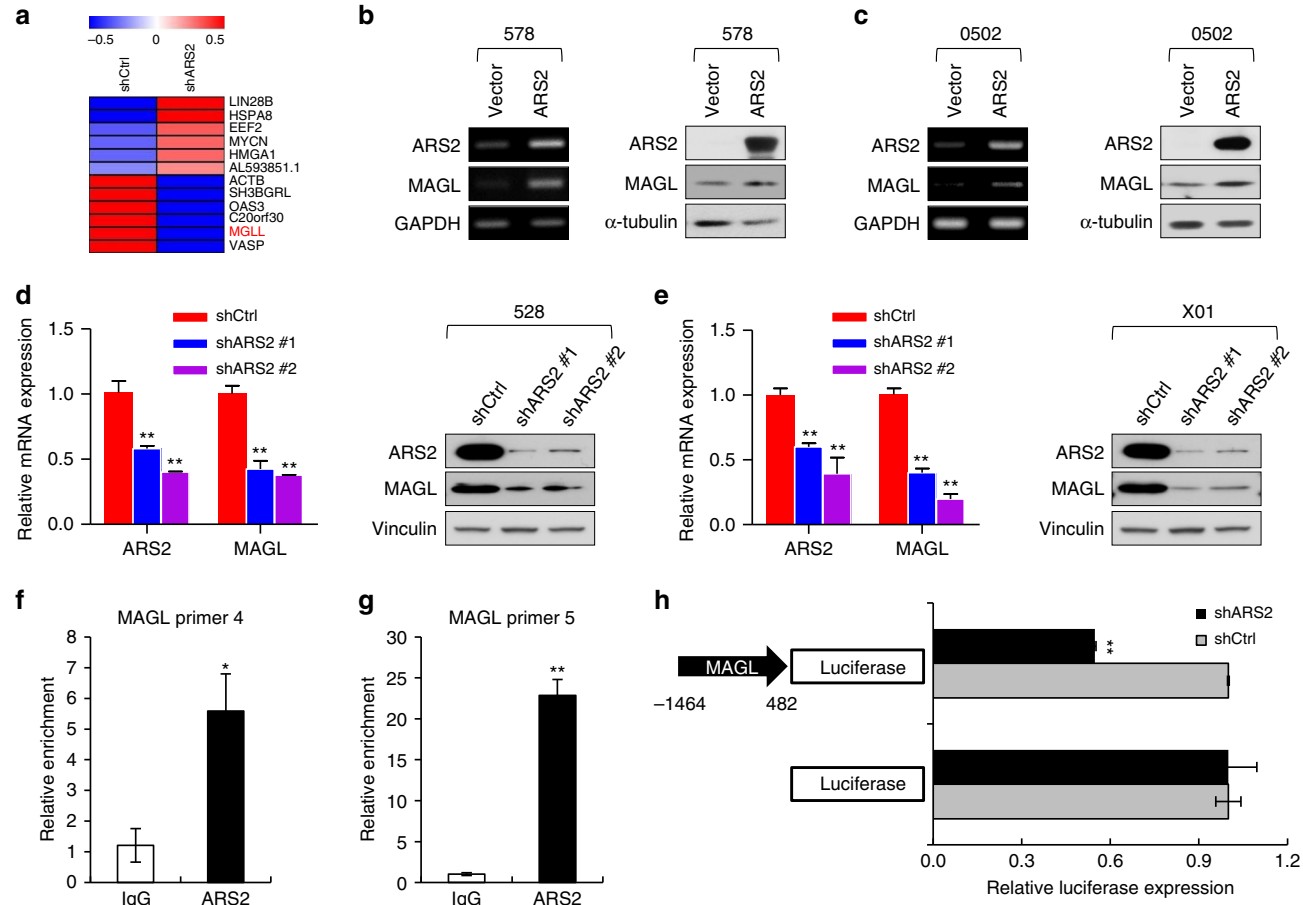

**Fig. 3 ARS2 directly activates transcription of *MGLL*. a** Heatmap analysis of gene expression in X01 cells infected with a shARS2-expressing lentiviral or shCtrl construct. **b**, **c** PCR (left) and immunoblot (IB) analysis (right) of ARS2 and MAGL in GSCs (578 and 0502 cells) infected with an ARS2-expressing lentiviral or vector construct. GAPDH or α-tubulin was used as a loading control. **d**, **e** qRT-PCR (left) and IB analysis (right) of ARS2 and MAGL in GSCs (528 and X01 cells) infected with a shARS2-expressing lentiviral or shCtrl construct. Vinculin was used as a loading control. All error bars represent mean ± SEM ($n = 3$). *$P < 0.05$, **$P < 0.01$, $t$-test. **f**, **g** Immunoprecipitated chromatin was analyzed by qPCR using 3- and 4-specific primers on the promoter region of the *MAGL* gene. All error bars represent mean ± SEM ($n = 3$). *$P < 0.05$, **$P < 0.01$, $t$-test. **h** Luciferase reporter assays of MAGL activity in GSCs (X01 cells) infected with a shARS2–expressing lentiviral or shCtrl construct. All error bars represent mean ± SEM ($n = 3$). **$P < 0.01$, $t$-test.

**MAGL roles in the self-renewal and tumorigenicity of GSCs.** We next asked whether MAGL itself regulates the self-renewal ability and tumorigenicity of GSCs. Sphere formation was remarkably increased in MAGL-overexpressing GSC 578 and 0502 cell lines (Fig. 4a–d). Conversely, deletion of MAGL in 528 and X01 cells completely abrogated sphere-forming ability (Fig. 4e–h). Moreover, MAGL knockdown in 528 and X01 cells induced expression of the astrocytic differentiation marker, GFAP, but decreased levels of the stemness marker, Nestin (Supplementary Fig. 4a, b). These loss- and gain-of-function analyses of MAGL demonstrated that the functional effects of modulating MAGL expression on GSC self-renewal are similar to those of modulating ARS2 expression. To examine the effects of ARS2-regulated MAGL expression on the recovery of GSC self-renewal capacity, we knocked down ARS2 and then over-expressed MAGL in GSC X01 cells. shRNA-mediated ARS2 knockdown markedly lessened self-renewal (Fig. 4i, j), whereas overexpression of MAGL in GSC X01 cells resulted in the highest degree of self-renewal detected (Fig. 4i, j). Notably, over-expression of MAGL in ARS2-knockdown X01 cells restored self-renewal ability, producing a degree of stemness virtually identical to that of controls (Fig. 4i, j).

To address the effects of MAGL in vivo, we orthotopically injected nude mice with control X01 cells or each of two different MAGL-knockdown X01 lines. Tumors were reduced in size, or were nonexistent, in MAGL-knockdown xenografts (Supplementary Fig. 4c). Xenografted mice injected with MAGL-depleted X01 cells survived significantly longer periods than mice injected with control X01 cells (Fig. 4k). These findings demonstrate that MAGL is capable of modulating the characteristics of GSCs, especially self-renewal and tumorigenicity.

**MAGL modulates self-renewal through PGE₂/pLRP6/β-catenin.** Next, we examined downstream elements in the MAGL pathway involved in regulating GSC self-renewal. MAGL hydrolyzes monoacylglycerols to produce FFAs and regulates the enzymatic conversion of AA to PGE₂[17]. To address the role of MAGL in GSCs, we first examined whether ARS2 depletion in GSCs modulates FFA levels by directly suppressing expression of MAGL, as previously reported in other cancers[17]. Ten naturally occurring FFAs were analyzed, and their number of carbons and degree of saturation were determined. No significant changes in the levels of any type of FFA were detected regardless of ARS2-knockdown status (Supplementary Fig. 5a), suggesting that MAGL in the brain, especially in GSCs, does not modulate FFAs, but instead controls production of PGE₂ in response to targeting by the upstream factor, ARS2.

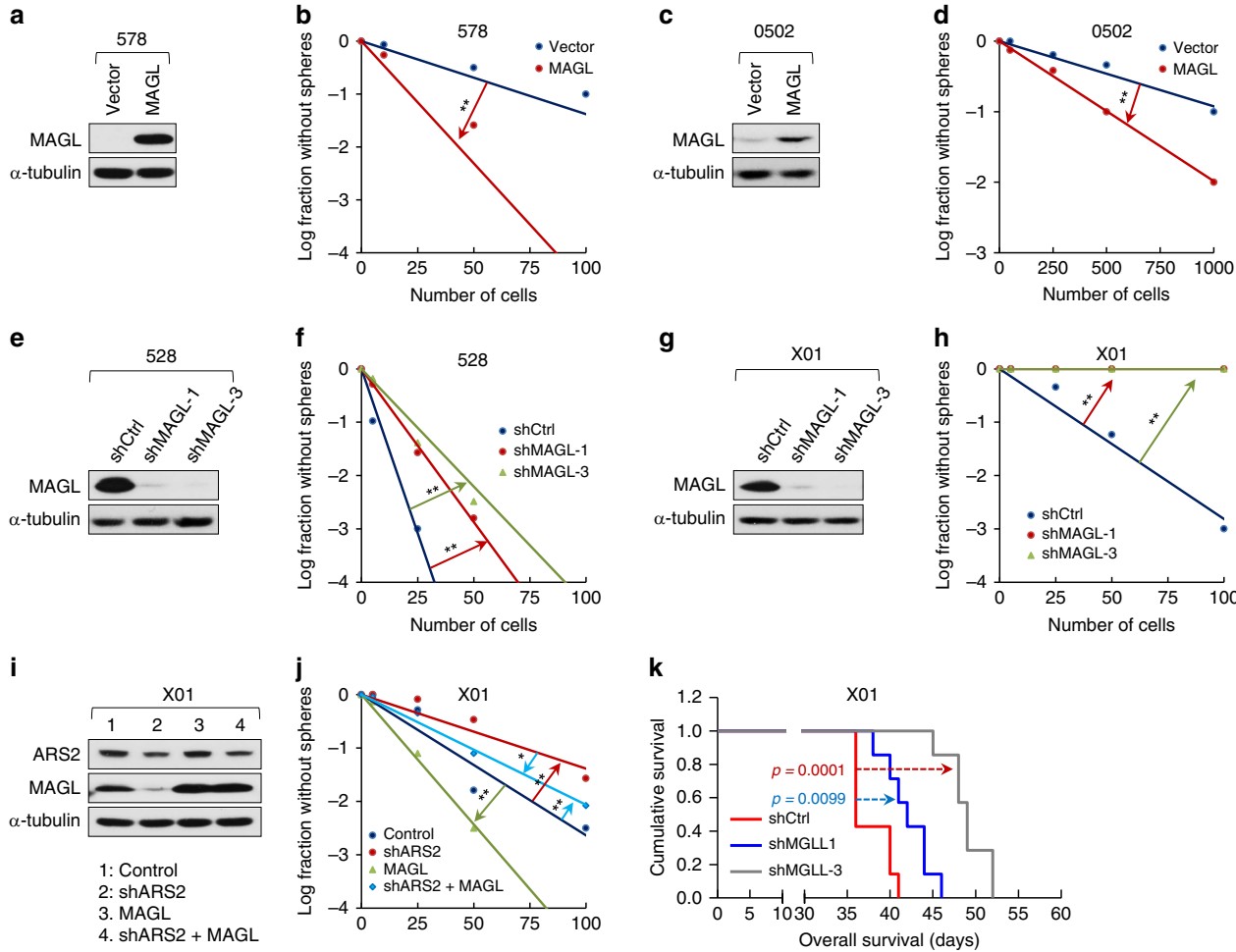

**Fig. 4 MAGL regulates the self-renewal and tumorigenesis of GSCs. a** Immunoblot (IB) analysis of MAGL in GSCs (578 cells) infected with a MAGL-expressing lentiviral or vector construct. α-tubulin was used as a loading control. **b** Limiting dilution assays (LDAs), performed in 578-vector and 578-MAGL cells. ($n = 24$, **$P < 0.01$, $t$-test). **c** IB analysis of MAGL in GSCs (0502 cells) infected with a MAGL-expressing lentiviral or vector construct. α-tubulin was used as a loading control. **d** LDAs, performed in 0502- vector and 0502-MAGL cells. ($n = 24$, **$P < 0.01$, $t$-test). **e** IB analysis of MAGL in GSCs (528 cells) infected with a shMAGL-expressing lentiviral or shCtrl construct. α-tubulin was used as a loading control. **f** LDAs, performed in 528-shCtrl, 528-shMAGL-1, and 528-shMAGL-3 cells. ($n = 24$, **$P < 0.01$, $t$-test). **g** IB analysis of MAGL in GSCs (X01 cells) infected with a shMAGL-expressing lentiviral or shCtrl construct. α-tubulin was used as a loading control. **h** LDAs, performed in X01-shCtrl, X01-shMAGL-1, and X01-shMAGL-3 cells. The well-formed sphere was counted at day 16. ($n = 24$, **$P < 0.01$, $t$-test). **i** IB analysis of ARS2 and MAGL in GSCs (X01 cells) infected with a shARS2- or MAGL-expressing lentiviral construct, both shARS2- and MAGL-expressing lentiviral constructs, or a shCtrl construct. **j** LDAs, performed in GSCs (X01 cells) infected with a shARS2- or MAGL-expressing lentiviral construct, both shARS2- and MAGL-expressing lentiviral constructs, or a control construct. The well-formed sphere was counted at day 7. ($n = 24$, *$P < 0.05$, **$P < 0.01$, $t$-test). **k** Kaplan–Meier survival analysis of mice implanted with X01 cells infected with shMAGL-expressing lentiviral or shCtrl construct ($n = 7$; $2 \times 10^4$ cells injected per mouse, log-rank test). Median survival of the orthotopic mice injected with X01-shCtrl, X01-shMGLL-1, or X01-shMGLL-3 was 36 days, 42 days, and 49 days, respectively.

Previous studies have suggested that PGE$_2$–stimulated β-catenin accumulation and the activation stimulates stemness in leukemia stem cells, growth of colon cancer cells, and progression of glioma[24–26]. Therefore, we selectively knocked down MAGL expression using MAGL-targeting shRNA (shMAGL) and subsequently assessed PGE$_2$ levels and β-catenin expression in GSCs (Fig. 5a, b and Supplementary Fig. 5b–e). Knockdown of MAGL simultaneously decreased PGE$_2$ production and β-catenin accumulation in GSCs (Fig. 5a, b and Supplementary Fig. 5b–e). Since β-catenin expression is closely related with its role in transcriptional activation, we determined whether ARS2 or MAGL affects theβ-catenin expression. We measured β-catenin protein levels in fractionated nuclear or cytosolic lysates after ARS2/MAGL knockdown. The successful fractionation of nuclear and cytosol proteins was validated with Lamin B and β-actin, respectively.

As expected, MAGL or ARS2 knockdown in 528 and X01 cells reduced β-catenin protein levels in nuclear fraction (Fig. 5c, d and Supplementary Fig. 5f, g). Moreover, treatment with PGE$_2$ increased β-catenin expression in a concentration-dependent manner in association with enhanced LRP6 phosphorylation (Fig. 5e). The sphere-forming ability of GSCs was also increased in a concentration-dependent manner by treatment with PGE$_2$ (Supplementary Fig. 5h), an effect that was blocked by the specific inhibitor of TCF/β-catenin-mediated transcription, ICG-001 (Fig. 5f, g)[27]. Treatment of GSC 528 and X01 cells with ICG-001 significantly reduced GSC sphere-forming ability through downregulation of TCF/β-catenin-mediated cyclin D1 and c-Myc transcription (Fig. 5h, i and Supplementary Fig. 5i, j). Collectively, these findings strongly suggest that MAGL regulates GSC self-renewal through PGE$_2$/pLRP6/β-catenin signaling.

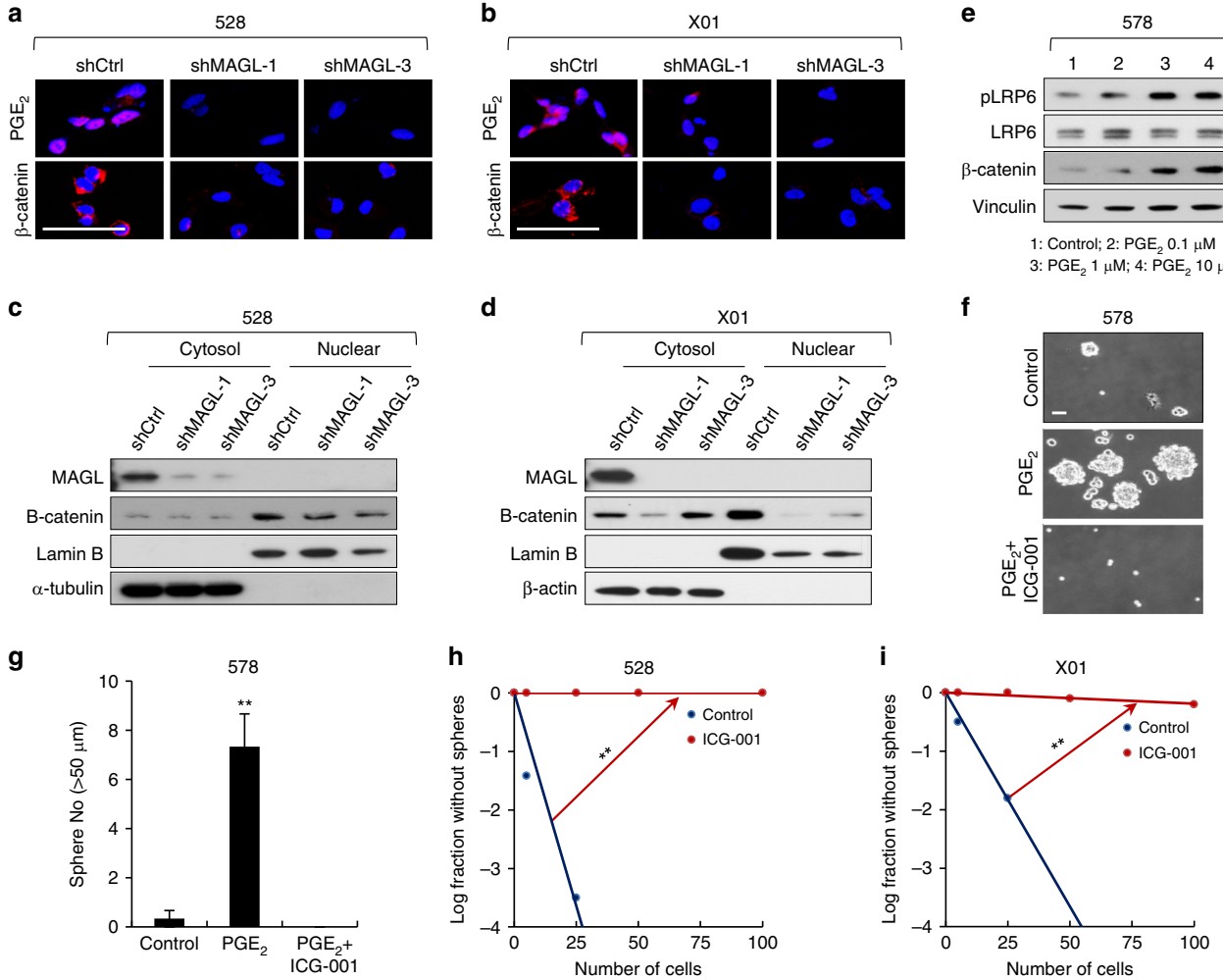

**Fig. 5 MAGL modulates GSC self-renewal by regulating PGE$_2$. a, b** Representative ICC images of PGE$_2$ and β-catenin in GSCs (528 and X01 cells) infected with a shMAGL-expressing lentiviral or shCtrl construct. Scale bar, 50 μm. **c, d** Immunoblot (IB) analysis of MAGL and β-catenin in fractionated nuclear or cytosolic lysates from 528 cells and X01 cells infected with a shMAGL-expressing lentiviral or shCtrl construct. Lamin B and β-actin were used as markers for nucleus and cytoplasm, respectively. **e** IB analysis of β-catenin in 578 cells treated with different concentrations of PGE$_2$ (0.1, 1, and 10 μM). Vinculin was used as a loading control. **f, g** Sphere-formation assays performed using GSCs (578 cells) treated with PGE$_2$ alone, PGE$_2$ and ICG-001 (10 μM), or vehicle. Images are representative of three independent experiments. Scale bar, 50 μm. The graph shows the average number of spheres >20 μm in diameter. All error bars represent mean ± SEM (n = 3). **P < 0.01, t-test. **h, i** Limiting dilution assays (LDAs) performed using GSCs (528 and X01 cells) treated with ICG-001 (10 μM) or vehicle. (n = 24, **P < 0.01, t-test.)

**ARS2/MAGL induces M2-like polarization of TAMs by PGE$_2$.** PGE$_2$, regulated by MAGL, is one of the most important factors in the brain that facilitates neuroinflammation[18]. Accordingly, we next examined the impact of downregulating ARS2 or MAGL on macrophage density in GSC-derived xenografts. In these xenograft mouse models, ARS2 or MAGL knockdown decreased staining for Iba-1, a marker of TAMs[28,29], suggesting inflammatory signaling associated with ARS2 or MAGL potentially regulates the tumorigenicity of GSCs (Supplementary Fig. 6a, b). Importantly, a number of studies have suggested that PGE$_2$ induces TAM polarization toward M2-like properties[30–34]. Hence, we explored whether PGE$_2$ induces subtype-specific TAM polarization in vitro using bone marrow-derived (BMDMs) and peritoneal cavity-extracted macrophages. It has previously been reported that these two subtypes of TAMs acquire opposite functions in relation to cancer: the M1-like subtype protects against cancer by suppressing angiogenesis, whereas the M2-like subtype is more likely to worsen cancer prognosis through enhanced invasion and tumor growth[9]. Accordingly, we compared the degree of induction of TAM polarization into M1 or M2 macrophages by treating with lipopolysaccharide (LPS) or interleukin (IL)−4, respectively, or with PGE$_2$. Treatment with PGE$_2$ induced expression of the M2-like macrophage markers, CD206 (MRC1), CD163, and ARG1 (arginase-1), to an extent comparable to that induced by IL4 (Fig. 6a, b and Supplementary Fig. 6c–e)[35,36]. Expression of Krupple-like factor 4 (KLF4), a key transcription factor that regulates expression of M2-like TAM genes[31], was significantly increased by IL4 or PGE$_2$ treatment (Fig. 6a and Supplementary Fig. 6c). In contrast, treatment of macrophages with PGE$_2$ reduced the expression of tumor necrosis factor (TNF)-α and CD86, markers of M1-like macrophages (Fig. 6a, c and Supplementary Fig. 6c, e, f)[31,35]. To further demonstrate PGE$_2$-induced M2-like TAM polarization, we sorted CD11b and F4/80 double positive TAMs from mouse xenograft model, regarded as more relevant to real TAMs, extracted from ARS2-knockdowned orthotopic xenograft mouse, and further confirmed the effect of PGE$_2$ on inducing M2-like polarization in the sorted TAMs. PGE$_2$ treatment on the FACS-sorted TAM from shARS2 and shMAGL tumors increased M2-like TAM marker CD206 expression, which supported our theory of ARS2/MAGL

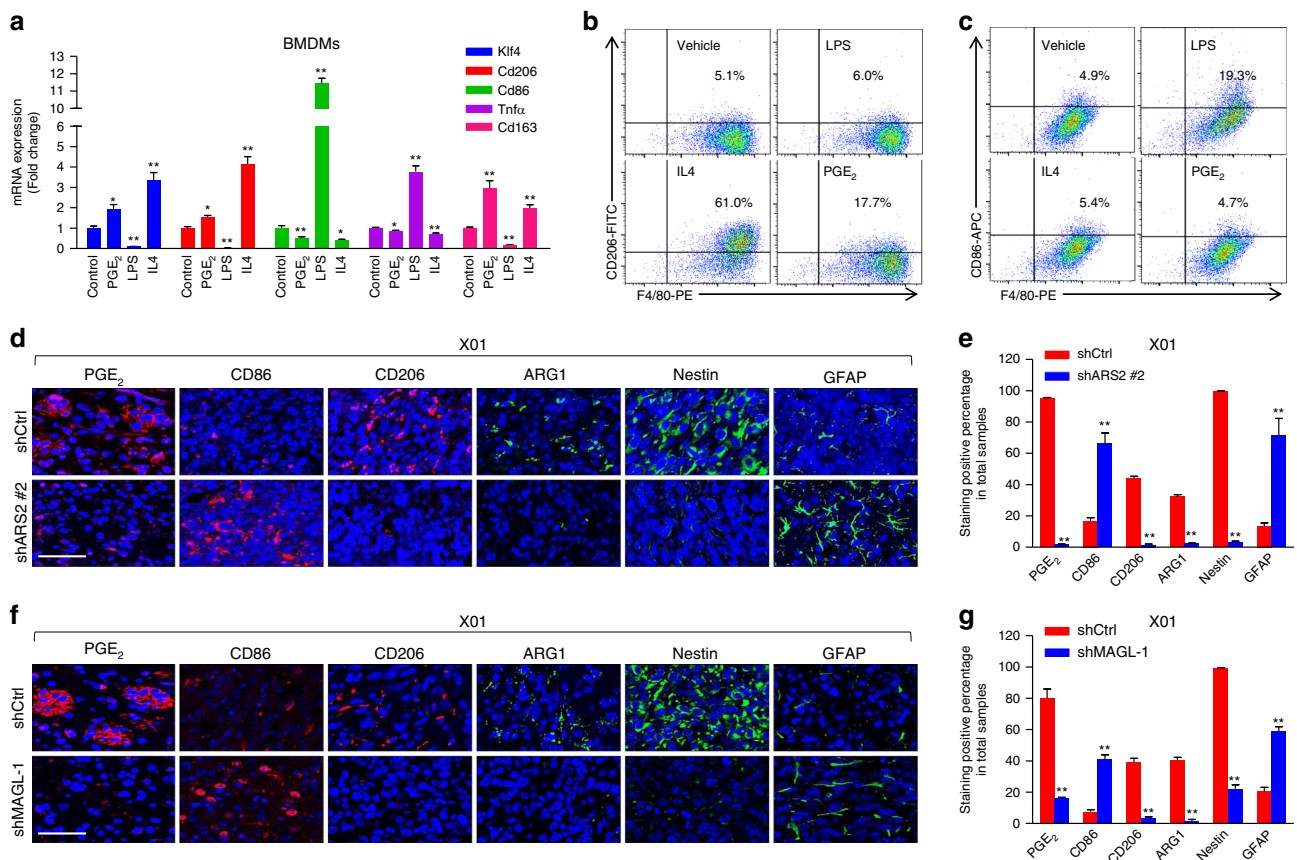

**Fig. 6 The ARS2/MAGL module regulates tumorigenesis through PGE$_2$–mediated induction of M2-like TAMs. a** qRT-PCR analysis of Klf4, Cd206 (Mrc1), Tnfα, and Cd163 in bone marrow derived-macrophages (BMDMs) after induction with PGE$_2$ (10 μM), LPS or IL4, or treatment with control. All error bars represent mean ± SEM ($n = 3$). *$P < 0.05$, **$P < 0.01$. **b, c** Expression of CD86 (**b**) and CD206 (**c**) in BMDMs treated with PGE$_2$ (10 μM), LPS, IL4, or vehicle control, and in negative BMDM control by flow cytometry. F4/80 was co-stained for macrophagic population verification. **d, e** Representative IF images (**d**) and corresponding quantification (**e**) of PGE$_2$, CD86, CD206, ARG1, Nestin, and GFAP in brain tissues from X01-shCtrl and X01-shARS2 #2 orthotopic xenograft mouse model. Scale bar, 50 μm. All error bars represent mean ± SEM ($n = 3$). **$P < 0.01$, $t$-test. **f, g** Representative IF images (**f**) and corresponding quantification (**g**) of PGE$_2$, CD86, CD206, ARG1, Nestin, and GFAP in brain tissues from X01-shCtrl and X01-shMAGL-1 orthotopic xenograft mouse model. Scale bar, 50 μm. All error bars represent mean ± SEM ($n = 3$). *$P < 0.05$, **$P < 0.01$, $t$-test.

modulating M2-like TAMs via PGE$_2$ (Supplementary Fig. 6g). Moreover, when we cultured GSC 578 cells in conditioned media (CM) from peritoneal macrophages which were inducted by LPS or IL4, respectively, IL4-induced M2-like-TAM CM increased expression of Nestin and decreased GFAP (Supplementary Fig. 6h). To further delineate the interaction between TAMs and GSC stemness, we hypothesized that M2-like TAMs secretes cytokines which likely promote GSC stemness. Here, we conducted cytokine arrays in macrophage after PGE$_2$ treatment (Supplementary Fig. 6i, j). PGE$_2$ enhanced the expressions of Lipocalin 2, Serpin E1, G-CSF, HGF, VEGF, and IL6, which was confirmed by real-time PCR (Supplementary Fig. 6i). Among the six genes, four of them (Lipocalin 2, HGF, VEGF, and IL6) were significantly upregulated by PGE$_2$ treatment (Supplementary Fig. 6i). Collectively, these results confirm that PGE$_2$ promotes M2-like TAM polarization which enhances GSC stemness by secretion of cytokines such as Lipocalin 2, HGF, VEGF, and IL6.

Next, we examined PGE$_2$ in GSCs upon the expression status of ARS2 and MAGL using an immunofluorescence approach (Fig. 6d–g). PGE$_2$ (red) was clearly detected in images of immune-stained tissue from mice injected with control X01 cells. As expected, PGE$_2$ was undetectable in tissues from mice injected with ARS2-knockdown (Fig. 6d, e) or MAGL-knockdown (Fig. 6f, g) GSCs. We also investigated M1- and M2-like subtypes of infiltrating TAMs by marker expression. The M2-like TAM

expression markers, CD206 and ARG1, were rarely detected in GBM tumors formed from GSCs virally infected with shARS2; instead, these tumors showed upregulation of the M1 marker CD86 (Fig. 6d, e). A similar increase in CD86 expression and decrease in CD206 and ARG1 expression and PGE$_2$ production was observed in GSCs infected with shMAGL (Fig. 6f, g). Therefore, our data support the conclusion that M1-like TAM polarization is increased by ARS2 or MAGL knockdown. Importantly, expression of the stemness marker Nestin was decreased and expression of the differentiation marker GFAP was increased in tumor tissues formed from ARS2- or MAGL-knockdown GSCs (Fig. 6d–g).

While our in vitro and in vivo data strongly supported the ability of ARS2 to concurrently drive glioma stemness and polarization into M2 macrophages, the complete elucidation of the details of ARS2 driven GSC-macrophages cross-talk will recessively require the use of highly specialized macrophages depleted immunodeficient mice.

Taken together, our data suggest that the elevated expression of MAGL in the brain controls neuroinflammation, as evidenced by the production of PGE$_2$, and further regulates GSC self-renewal and M2-like TAM polarization.

**Blocking MAGL impairs the self-renewal and tumorigenicity.** To further investigate efficacy of MAGL blockade in regulating the stemness characteristics of GSCs, we employed an in vitro

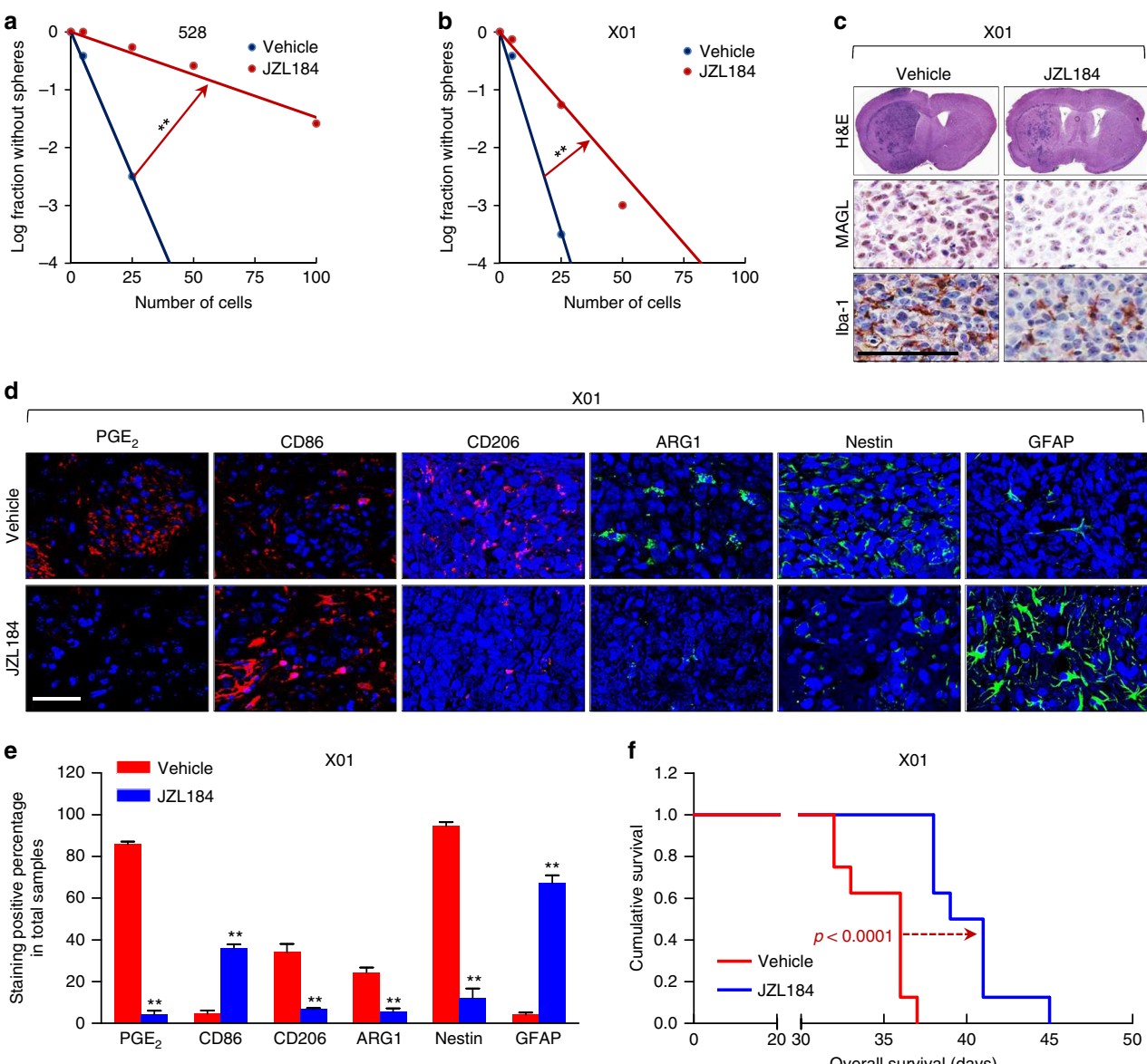

**Fig. 7 JZL184 inhibits GSC self-renewal and tumorigenicity. a, b** Limiting dilution assays (LDAs) performed using 528 cells (B) and X01 cells (C) treated with JZL184 (1 μM). (n = 24). **c** H&E staining of the whole brain and immunohistochemical (IHC) analysis of MAGL and Iba-1 in a JZL184-treated orthotopic xenograft mouse model. Scale bar, 100 μm. The sample is extracted at 32 days after cell injection. **d, e** Representative IF images (**d**) and corresponding quantification (**e**) of PGE$_2$, CD86, CD206, ARG1, Nestin, and GFAP in a JZL184-treated orthotopic xenograft mouse model. Scale bar, 50 μm. All error bars represent mean ± SEM (n = 3). *P < 0.05, **P < 0.01, t-test. **f** Kaplan–Meier survival analysis of mice implanted with X01 cells treated with JZL184 or vehicle (n = 8; 2.5 × 10$^4$ cells injected per mouse, log-rank test). Median survival of the mice treated with vehicle, or JZL184 was 36 days, and 40 days, respectively.

pharmacological approach using the MAGL-specific inhibitor, JZL184[15]. JZL184 at a concentration as low as 1 μM eliminated detectable MAGL hydrolase activity in the GSC cell lines, 528 and X01 (Supplementary Fig. 7a). This abrogation of MAGL expression was associated with a remarkable decrease in the renewal capacity of both GSC cell lines, as shown by the results of limiting dilution assays (Fig. 7a, b).

Next, we orthotopically implanted GSC X01 cells into the brains of nude mice, and then treated mice orally with JZL184 or vehicle every day. Treatment with JZL184 decreased tumor mass; immunohistochemical staining further showed that JZL184 decreased MAGL levels and the number of Iba-1–expressing cells (Fig. 7c). Notably, administration of JZL184 was sufficient to suppress MAGL expression in GSCs and infiltration of TAMs (Fig. 7c).

Production of PGE$_2$, the final manifestation of MAGL-induced neuroinflammation, was also extinguished by treatment with JZL184, as demonstrated by immunofluorescence (Fig. 7d). M1-like TAM polarization, represented by the marker CD86, was increased in X01 cells by treatment with JZL184, whereas M2-like TAMs, marked by CD206 and ARG1 expression, exhibited an opposite response to JZL184 administration (Fig. 7d, e). We confirmed that treatment with JZL184 downregulated the stemness marker Nestin, and upregulated GFAP (Fig. 7d, e). Finally, we assessed the survival of xenograft mice following JZL184 treatment. These experiments revealed that survival was significantly longer for JZL184-treated xenograft mice compared with vehicle-treated controls (Fig. 7f).

Furthermore, we blocked PGE$_2$ production by treating PGE$_2$ synthesis inhibitor, celecoxib in subcutaneous mouse xenograft

model. As shown in Supplementary Fig. 7b–d, celecoxib significantly decreased tumor size and volume, modulated M1- and M2-like TAMs, and reduced expression of stemness markers same as the effect of JZL184. These beneficial effects of JZL184 or celecoxib treatment indicate that pharmacological inhibition of MAGL or COX2 to block PGE$_2$ production suppresses the self-renewal and tumorigenic capacity of GSCs and promotes M1-like polarization of TAMs.

**JZL184 downregulated M2-Like TAM signature for patients survival.** To elucidate the global transcriptional change in tumor-resident macrophages, we established subcutaneous mouse models of GSC X01 cells. Then, CD11b$^+$ macrophages were isolated by fluorescence-activated cell sorting (FACS) from JZL184-treated mice. With constructed mouse models, we performed RNA-Seq to compare the gene set between the experimental mice and respective control X01 xenograft mice (Supplementary Fig. 8a). Signature genes downregulated by JZL184 in RNA-Seq were positively correlated with M2 TAM gene signatures[37] (Fig. 8a). We next examined the expression of the signature genes downregulated in the previously sorted CD11b$^+$ macrophages, and also in the newly sorted CD11b and F4/80 double positive TAMs from subcutaneous mouse xenograft models. JZL184 treatment decreased M2-like TAM genes expression, significantly (Supplementary Fig. 8b, c). So, we referred the gene signature downregulated by treatment of JZL184 as M2-like TAM signature. Since the role of infiltrating M2 macrophage in GBM has been implicated in mesenchymal

(MES) subtype patients and associated with poor prognosis of GBM[38], we examined possible relationship between M2-like macrophage enrichment and MES subtype. Interestingly, we observed strong expression of CD44 (a MES subtype marker) and ARG1 in edge region of control but not in JZL184 treated-orthotopic xenograft mouse model (Fig. 8b and Supplementary Fig. 8d, e). These results suggest that pharmacological inhibition of MAGL blocks MES subtype change which is triggered by infiltrating M2-like macrophage. Furthermore, we also analyzed M2-like signature in GBM patients and identified that M2-like signature was enriched in mesenchymal subtype patients, while patients with lower expression of the signature harbored non-mesenchymal subtypes (Fig. 8c). Higher expression of M2-like TAM signature predicted significantly shorter survival in 161 TCGA patient pools (Fig. 8d).

Gene set enrichment analysis (GSEA) further indicated that inhibition of MAGL by JZL184 reduced stemness and invasiveness signatures (Fig. 8e and Supplementary Fig. 8f). Moreover, GSEA analysis revealed that JZL184 treatment increased T lymphocyte associated gene sets (Supplementary Fig. 8g, h). So, we designed combination treatment of JZL184 and anti-PD-1 monoclonal antibody in syngeneic mouse model, which showed synergistic effect to improve the median survival (Fig. 8f). Our results of the successful combination of JZL184 to the immune checkpoint therapy are suggestive of a new promising therapeutic approach. In summary, the above data indicate that pharmacological inhibition of MAGL by JZL184 reduces expression of a M2-like TAM gene signature and is a hallmark of tumor aggressiveness of GBM patients.

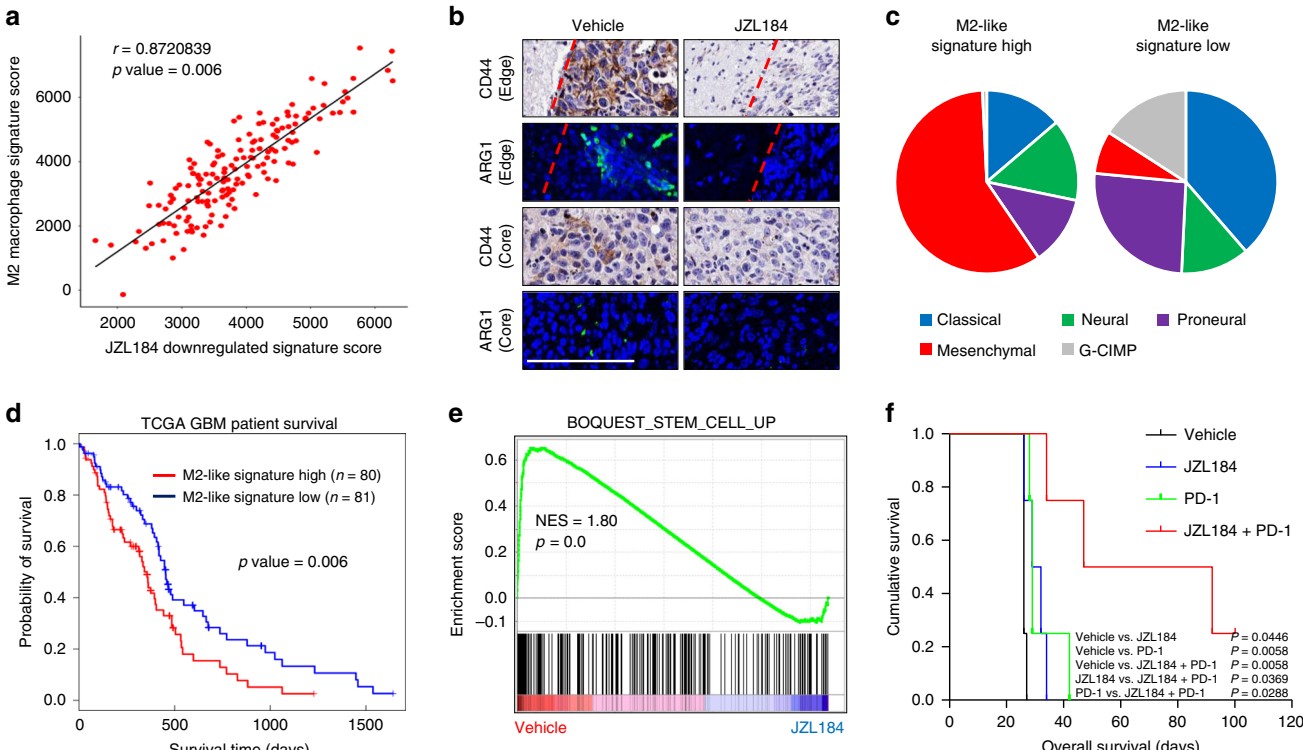

**Fig. 8 JZL184 inhibits M2-like TAMs signature and associated with patients survival. a** Correlation dot-plot of JZL184 downregulated gene signature and M2 gene signature of TAMs from vehicle vs. JZL184-treated subcutaneous mouse model. **b** Representative IHC and IF images of CD44 and ARG1 in edge or core tumor regions of brain slices treated with JZL184 or vehicle. Scale bar, 100 μm. **c** Pie chart of ratio of GBM subtypes according to high and low expression of M2-like gene signature. **d** Kaplan–Meier survival plots for GBM patients in the TCGA data set according to high (top 50%) and low (bottom 50%) M2-like signature expression. **e** Gene Set Enrichment Analysis (GSEA) plot for stemness gene signature in comparison of TAMs from vehicle vs. JZL184-treated subcutaneous mouse model. **f** Kaplan–Meier survival plots for the orthotopic syngeneic mouse model (n = 4). Median survival of the orthotopic syngeneic mice treated with Vehicle, PD-1, JZL184, or PD-1+JZL184 was 26 days, 29 days, 30.5 days, and 69.5 days, respectively.

## Discussion

In the current study, we demonstrate that ARS2 is a regulator of stem-cell identity in GBM, showing that it upregulates MAGL by directly mediating the transcription of the corresponding gene *MGLL* in GSCs. MAGL, in turn, enhances self-renewal ability and tumorigenic activation through PGE$_2$–mediated β-catenin accumulation and M2-like TAM polarization. Moreover, we provide the first demonstration that pharmacological inhibition of MAGL effectively suppresses GSC self-renewal and tumorigenicity—the major therapeutic challenges in GBM.

Most research on ARS2 to date has focused its significant regulatory roles in early development and proliferation[12,13], studies that have shown that genetic deletion of *Ars2* results in embryonic lethality. In its capacity as a component of the nuclear cap-binding complex, ARS2 acts through miRNA biogenesis to induce cell proliferation[39]. Although this transcriptional role of ARS2 in development is well represented in the literature, the potential relationship of ARS2 with cancer, particularly with GSCs, has received relatively little research attention. Our findings indicate that ARS2 expression is correlated with poor prognosis of glioma patients and, importantly, promotes the tumorigenicity of GSCs through direct transcriptional induction of *MGLL* in the brain. These findings demonstrate a new molecular mechanism for ARS2, showing that it is a critical transcription activator of MAGL in GSC self-renewal and tumorigenic processes. Furthermore, by enhancing our understanding of the transcriptional network in tumor-initiating GSCs, the results of this study will aid in establishing tailored targeting strategies.

MAGL, a membrane-associated enzyme in the cytosol that catalyzes the release of FFAs from lipid chains, is highly expressed in advanced ovarian tumors and was recently reported to be essential for remodeling of the lipid network[15–17]. In the context of cancer, it has been shown that MAGL expression is elevated in aggressive human cancers and primary tumors, including breast, ovarian, and melanoma cancers[17,22,23]. Another function of MAGL suggested by a recent study is as a crucial modulator of PGE$_2$, which is capable of directly sustaining β-catenin accumulation[26]. Likewise, our results demonstrate that MAGL acts through regulation of PGE$_2$ production, and not FFA-associated lipid modulation, to influence GSC self-renewal and tumorigenicity. Previous studies have suggested that PGE$_2$ simulates β-catenin accumulation, which is an important factor in activation of leukemia stem cells and glioma progression[24–26]. As expected in light of these findings, our examination of the mechanistic role of PGE$_2$ in β-catenin signaling in GSCs reveals that MAGL-mediated PGE$_2$ production induces β-catenin accumulation in GSCs. Moreover, treatment with PGE$_2$ increased β-catenin expression in a concentration-dependent manner. These observations reinforce our experimental finding that MAGL expression is associated with the maintenance of GSC characteristics.

Although downstream effects of MAGL on the lipolytic network have been extensively investigated, questions related to upstream regulation of MAGL activity in cancer cells have remained unresolved[40]. In the current study, we propose that ARS2, an important transcription factor in GSCs, exerts control over MAGL by regulating the expression of its corresponding gene, *MGLL*, and that the ARS2-MAGL functional module plays an important role in neuroinflammation. Our ChIP assays using multiple primers covering different regions of the *MGLL* promoter demonstrated that the *MGLL* gene is a direct target of ARS2, further supporting the functional association between ARS2 and MAGL. Our study is also the first to show that MAGL maintains the characteristics of GSCs.

An additional interesting relationship explored here is the association between TAMs and GBM. GBM is a complex disease, and TAMs further exacerbate this complexity[9]. On the basis of correlation studies relating the survival of glioma patients with the expression of TAMs, previous investigators have classified TAMs into M1 and M2 subtypes according to their characteristics and specific markers[9]. Although this classification scheme is not absolute, each subtype generally exhibits a phenotype opposite that of the other: the M2-like subtype is correlated with high-risk glioma patients and tumor invasion, whereas the M1-like subtype is associated with suppression of angiogenesis and tumor growth—characteristics that may offer promising therapeutic advantages[9,41]. In the current study, we utilized an immunofluorescence approach to visualize CD86, a marker of M1-like TAM polarization, and CD206 and ARG1, markers of M2-like polarization. Overall, knockdown of ARS2 or MAGL expression decreased neuroinflammation, but different trends in ARS2 and MAGL expression were observed in relation to the phenotypes of TAMs: whereas ARS2 and MAGL expression in GSCs were both correlated with upregulation of the M2-like phenotype, knockdown of either factor led to an increase in the M1-like polarization. Therefore, this study confirms previously reported functions of both types of TAMs in glioma.

The MAGL-selective inhibitor JZL184 virtually eliminated MAGL expression in GSCs. Already known for its high selectivity for MAGL, JZL184 treatment clearly suppressed MAGL expression both in vitro and in vivo, in association with a significant reduction in the self-renewal capacity of GSCs and suppression of neuroinflammation. It should be noted that previous studies in mice have shown that treatment with JZL184 is accompanied by side effects related to cannabinoid receptor 1-dependent signaling, including analgesia, hypothermia, and/or gastrointestinal bleeding[42,43]. Although our data support an important role for the ARS2-dependent MAGL mechanism described here, a comprehensive analysis of the safety profile of MAGL inhibitors as well as large-scale, patient-oriented studies designed to assess effectiveness should be conducted to confirm the clinical viability of targeting this mechanism. Such studies will hopefully facilitate translation of a future drug to treat aggressive GBM from bench to bedside.

In conclusion, we identified ARS2 as a new and important transcription factor that promotes the stem cell identity of GSCs through MAGL-mediated signaling and further showed that blockade of MAGL provides a promising therapeutic avenue for treating GBM.

## Methods

**Cell culture**. 293T cells were maintained in Dulbecco's modified Eagle's medium (DMEM) supplemented with 10% fetal bovine serum (HyClone). Patient-derived GBM stem cells (528, X01, 578, and 0502) were maintained in DMEM/F-12 supplemented with B27 (Invitrogen), EGF (10 ng/ml, R&D Systems), and bFGF (5 ng/ml, R&D Systems). PGE$_2$, ICG-001, and JZL184 was purchased from R&D systems, Peprotech, Tocris Bioscience, respectively. All cells were repeatedly screened for mycoplasma and maintained in culture for <6 months after receipt.

**Plasmids**. The lentiviral construct pLenti6-ARS2-FLAG, expressing C-terminally FLAG-tagged human ARS2, was generated by first amplifying ARS2 by PCR using pLenti6-ARS2 (Cosmogenetech, unpublished plasmid), as a template, and then inserting the amplified DNA fragment into *Bam*HI-*Xba*I-CIP-treated pcDNA3-cFLAG (J.H.K., unpublished plasmid) generating pcDNA3-ARS2-FLAG. The final pLenti6-ARS2-FLAG construct was prepared by assembling *Eco*RI-*Xho*I–treated pLenti6-MCS (J.H.K., unpublished plasmid) and pcDNA3-ARS2-FLAG via an In-Fusion reaction (Takara). The lentiviral construct, pLenti6-MAGL, was generated by assembling *Bam*HI-*Eco*RI-treated pLenti6-MCS and MAGL, PCR-amplified from a human brain cDNA library (Clontech). For construction of the lentiviral construct containing the *MGLL* promoter region (pGreen-MAGL-pro), a ~2-kb region of the *MGLL* moter, amplified by PCR using Huh7 cell genomic DNA as a template, was inserted into *Eco*RI-*Spe*I-digested pGreenFire1 using an In-Fusion

reaction (Takara). shRNA-expressing lentiviral constructs targeting ARS2 and MAGL were constructed by ligating annealed oligomers (see Supplementary Table 2) with AgeI-EcoRI-digested pLKO.1puro (Addgene). All oligomers were purchased from Macrogen (Seoul, Korea). All constructs were verified by DNA sequencing (Cosmogenetech).

**Lentivirus production and infection**. Lentiviruses were produced as previously reported[29,44]. Briefly, $3-4 \times 10^6$ 293T cells were plated on 100-mm culture dishes, incubated for 24 h, and then co-transfected with 4.5 µg of lentiviral constructs, 3 µg of psPAX2 (Addgene), and 1.5 µg of pMD2.G (Addgene) using 27 µl of Lipofectamine 2000 (Invitrogen). The medium was changed 6 h after later, and 48 h after transfection, medium containing lentivirus was harvested. Viral particles were concentrated and purified using a Lenti-X concentrator (Clontech). Cells were infected with lentivirus in the presence of 10 µg/ml polybrene.

**Reverse transcription-polymerase chain reaction**. Semi-quantitative and real-time reverse transcription-polymerase chain reaction (RT-PCR) was performed to determine mRNA levels as previously described[29,45,46]. Total RNA was isolated from cells using TRIzol reagent (Invitrogen) according to the manufacturer's instructions. Total RNA (1 µg) was used as a template to synthesize cDNA using M-MLV reverse transcriptase (Promega). Real-time RT-PCR analysis was performed on a LightCycler 480II real-time detection system (Roche) using Light-Cycler 480 SYBR Green I Master Mix (Roche). The expression levels of target genes were normalized to that of GAPDH. Semi-quantitative RT-PCR products were analyzed on 1% agarose gel. The PCR primers are shown as follows: ARS2, sense 5′-AAGCTGGATTTCCGGAGGCA-3′ and antisense 5′-CCTGCTCCAG GATGCGAAGA-3′; MAGL, sense 5′-CCGCAGAGCATTCCCTACCA-3′ and antisense 5′-GCTGCAACACATCCCTGACG-3′; GAPDH, sense 5′-GGAGTC CACTGGCGTCTTCAC-3′ and antisense 5′-GAGGCATTGCTGATGATCTT GAGG-3′; mKlf4, sense 5′-GCTCCTCTACAGACCGAGAAT-3′ and antisense 5′-AGCACAAACTTGCCCCATCAG-3′; mCd206, sense 5′-TACCTGAGCCCACA CCTGCT-3′ and antisense 5′-GCGCGTTGTCCATGGTTTCC-3′; mCd86, sense 5′-CCCGGATGGTGTGTGGCATA-3′ and antisense 5′-TCACAAGGAGGAGG GCCACA-3′; mTnfα, sense 5′-TGGCCAACGGCATGGATCTC-3′ and antisense 5′-GGGCAGCCTTGTCCCTTGAA-3′; and mCd163, sense 5′-CATGTCTCT-GAGGCTGACCA−3′ and antisense 5′-TGCACACGATCTACCCACAT−3′.

**Limiting dilution assays**. For in vitro limiting dilution assays, GSCs were plated at decreasing densities (100, 50, 25, and 5 cells/well) in 96-well plates containing DMEM/F-12 supplemented with B27, epidermal growth factor (EGF; 10 ng/ml), and basic fibroblast growth factor (bFGF; 5 ng/ml). Limiting dilution assay results were processed using ELDA (Extreme Limiting Dilution Analysis) software, available at http://bioinf.wehi.edu.au/software/elda/.

**Cell proliferation assays**. GSCs were plated at $10^3$ cells/well densities in 96-well plates containing DMEM/F-12 supplemented with B27, epidermal growth factor (EGF; 10 ng/ml), and basic fibroblast growth factor (bFGF; 5 ng/ml) for in vitro Proliferation assays. The luminescence of viable cells was detected using CellTiter-Glo Luminescent Cell Viability Assay Kit according to the protocol of the manufacture (Promega). CellTiter-Glo Luminescent Cell Viability Assay is a homogenous method of determining the number of viable cells in culture based on quantitation of the ATP present, which signals the presence of metabolically active cells. The luminescence signal was detected by SpectraMax L Microplate Reader (Molecular Device) according to the manufacturer's protocol. The result was analyzed and visualized using Microsoft excel 2010.

**Metabolite measurements**. Approximately $10^6$ cells were used for determination of FFAs. After removing medium and washing with 20 mL PBS, cells were fixed by adding 500 µL of methanol. Fixed cells were obtained from the culture dish using a cell scraper and collected in a glass tube; cells remaining in the culture dish were removed using an additional 500 µL of methanol and pooled with the initially collected cells. The glass tube was vortexed vigorously to completely lyse cells, after which HCl was added to a final concentration of 25 mM. Internal standard solution (50 µL of 0.1 mg/mL myristic-d27 acid) was added to the sample solutions and mixed well, after which samples were microcentrifuged for 10 min at $340 \times g$. Supernatants were collected in fresh tubes, and 3 mL of isooctane were added to each sample. After mixing well, samples were centrifuged for 15 min at $2000 \times g$ and the upper layer was collected. This extraction step was repeated two more times. The collected upper layer was dried using a vacuum centrifuge and stored at $-20$ °C until gas chromatography/mass spectrometry (GC/MS) analysis. Methyl esterification of FFAs was performed after reacting with $BCl_3$-MeOH at 60 °C for 30 min. All lipid standards, including internal standards, were purchased from Avanti-Polar Lipids and Sigma-Aldrich (St. Louis, MO, USA).

Fatty acid methyl esters were analyzed using an Agilent 7890/5975 GC/MSD system equipped with an HP-5 MS 30 m × 250 µm × 0.25 µm column (Agilent 19091S-433). Helium (99.999%) was used as carrier gas, and samples were run in scan mode, with application of a 5-min solvent delay. The initial temperature was 50 °C, and was raised to 120 °C at a rate of 10 °C/min and held for 2 min.

Thereafter, temperature was raised to 250 °C at a rate of 10 °C/min and was maintained at that temperature for 15 min. The GC column was cleaned between runs by heating to 300 °C. The extracted ion chromatogram corresponding to a specific fatty acid was used for quantitation.

**Immunoblot analysis**. Proteins were extracted with RIPA buffer with complete protease inhibitors (Roche), separated by electrophoresis, transferred to PVDF membranes (Millipore), and blocked with 5% skim milk (BD). Primary antibodies against ARS2 (GeneTex, GTX119872, 1:500, 101 kDa), MAGL (Santa Cruz Biotech, ab24701, 1:1000, 33 kDa), pLRP6 (Ser1490, Cell Signaling Technology, #2568, 1:1000, 200 kDa), LRP6 (C47E12, Cell Signaling Technology, #3395, 1:1000, 200 kDa), β-catenin (D10A8, Cell Signaling Technology, #8480, 1:1000, 92 kDa), Lamin B (C-20, Santa Cruz Biotech, sc-6216, 1:1000, 67 kDa), Cyclin D1 (A-12, Santa Cruz Biotech, sc-8396, 1:1000, 37 kDa), c-Myc (C-12, Santa Cruz Biotech, sc-398624, 1:1000, 67 kDa), Vinculin (H-10, Sigma-Aldrich, sc-25336, 1:1000, 116 kDa), and α-tubulin (TU-02, Santa Cruz Biotech, sc-8035, 1:1000, 55 kDa) were incubated overnight at 4 °C. Immunoreactive bands were visualized using peroxidase-labeled affinity purified secondary antibodies (KPL) and the Amersham ECL prime western blotting detection reagent (GE Healthcare). Some of the western blot images were obtained by C-DiGit® Blot Scanner (LI-COR Biosciences).

**Immunoblot assay for MAGL hydrolase activity**. GSCs ($1 \times 10^6$ cells) were plated in 6 cm dishes with GSC culture media. After 24 h, cells were treated with JZL184 (1 µM and 2 µM) or vehicle (DMSO) for 24 h and harvested for MAGL hydrolase activity measurements. Proteins were extracted with RIPA buffer with complete protease inhibitors (Roche), and 100 µg proteins were treated with 2 µM ActivX TAMRA-FP Serine Hydrolase Probes (Thermo Scientific) for 30 min at room temperature (100 µl total reaction volume). Reactions were quenched with relative volume of standard 5x SDS/PAGE loading buffer (reducing), separated by SDS/PAGE, and visualized in-gel with a Typhoon FLA 700 gel scanner (GE).

**Chromatin immunoprecipitation**. For each ChIP reaction, $\sim 1 \times 10^6 \times 01$ cells were crosslinked with 1% formaldehyde for 10 min at room temperature, and genomic DNA was fragmented into ~100–300 bp fragments by sonication according to the manufacturer's instructions (MAGnify Chromatin Immunoprecipitation System, Thermo-Fisher Scientific). DNA-bound ARS2 was immunoprecipitated using an ARS2-specific antibody (Genetex). The associated DNA was then purified and analyzed by qRT-PCR to detect specific DNA sequences within the MGLL promoter that were bound in vivo by ARS2 protein. An antibody against IgG (MAGnify Chromatin Immunoprecipitation System, Thermo-Fisher Scientific) was used as a nonspecific control and histone H3 antibody as positive ChIP grade control (Abcam, ab18521, 1:1000, 15kD).

**Immunocytochemical staining**. GSCs ($1.5 \times 10^4$ cells) were grown on 8-well chambered culture slides (Nunc). After 24 h, cells were fixed with 2% paraformaldehyde (PFA) and permeabilized by incubating with 0.25% Triton X-100 for 10 min at room temperature (RT). After permeabilization, GSCs were immunostained for the cancer stem cell markers, Nestin (BD Biosciences, 611658, 1:500) and GFAP (MP Biomedicals, 691102, 1:500), as ARS2 (Genetex, GTX119872, 1:500), $PGE_2$ (Abcam, ab2318, 1:100), and β-Catenin(D10A8, Cell Signaling Technology, #8480, 1:500), CD86 (EP1158Y, Abcam, ab53004, 1:500), CD206 (Abcam, ab64693, 1:500), ARG1 (E-2, Santa Cruz Biotech, sc-271430, 1:500) by incubating overnight at 4 °C in a humidified chamber with primary antibody, diluted for the working concentration with antibody diluent buffer (IHC World). Immunoreactive proteins were visualized with the appropriate Alexa Fluor 488- or Alexa Fluor 568-conjugated secondary antibody (Thermo-Fisher Scientific, 1:500). Nuclei were stained with 4′,6-diamidino-2-phenylindole (DAPI; Sigma, 1:50000). Fluorescence images were obtained using an LSM 780 confocal laser-scanning microscope (Carl Zeiss).

**In vivo study**. All animal experiments were conducted in accordance with protocols approved by the Institutional Animal Care and Use Committee of the National Cancer Center, Republic of Korea. Every animal was randomized by body weight before conducting experiments. For the orthotopic mouse model[29,47], cells were first resuspended in DMEM/F-12 supplemented with B27, EGF (10 ng/mL), and bFGF (5 ng/mL), and then transplanted into the left striatum of 5-week-old female BALB/c nude mice by stereotactic injection. The injection coordinates were 2.2 mm to the left of the midline and 0.2 mm posterior to the bregma at a depth of 3.5 mm. For syngeneic orthotopic model, GL261 cells were injected into 5-week-old female C57BL/6 mice. The tumors were extracted, pooled for each experimental group, and mechanically disaggregated using stainless steel operating scissors. The brain of each mouse was harvested and fixed in 4% PFA. JZL184 (30 mg/kg; Tocris Biosciences) was orally administered daily. PD-1 antibody (10 mg/kg; Bioxcell) was administered once a week for 9 weeks through an intraperitoneal injection. Survival was analyzed using GraphPad PRISM software (version 7; GraphPad PRISM, La Jolla, CA, USA).

**Isolation and activation of mouse peritoneal macrophages**. Three days after intraperitoneal injection of 1–5% thioglycolate into the abdominal cavity of a nude mouse, peritoneal macrophages were obtained under sterile conditions. The cells were harvested by washing the peritoneal cavity with cold PBS (Gibco), then centrifuged ($300 \times g$, 7 min) and resuspended in Dulbecco's modified Eagle's medium (DMEM) (Gibco) supplemented with 10% fetal bovine serum and 1% penicillin/streptomycin (P/S) (Sigma-Aldrich, St. Louis, MO, USA). The cells were allowed to adhere for 4 h and then were washed to remove non-adherent cells and cultured in DMEM supplemented with 1% P/S. Finally, the purified macrophages were activated by incubating for 48 h with LPS (10 ng/mL; Sigma-Aldrich) or IL-4 (10 ng/mL; PeproTech).

**Isolation and activation of mouse bone marrow-derived macrophages**. Bone marrow cells were collected from female BALB/c nude mice femurs and tibias, and cultured with 20 ng/ml recombinant M-CSF (BioLegend) for 5 days in a petri dish. On day 5, naïve BMDMs were stimulated with LPS (10 ng/mL; Sigma-Aldrich) or IL-4 (10 ng/mL; Selleckem) to generate the BMDM-M1 or BMDM-M2 macrophages, respectively, for 24 h.

**Flow cytometry**. For surface marker analysis, live cells were re-suspended in 0.1% BSA 1xPBS and stained with anti-mouse F4/80 (PE, 1:100, Invitrogen), CD11b (PerCP-Cyanine5.5, 1:100 Invitrogen), and CD86 (APC, 1:100, BioLegend) at 4 °C for 20 min. Cells were fixed and permeabilized (Cytofix/cytoperm, BD) for intracellular protein staining, then labeled with anti-mouse CD206 (FITC, 1:100, Bio-Legend). Data were acquired by the BD LSRFortessa and analyzed with FACS Diva and FlowJo software. Cell sorting was performed by BD Cell Sorting FACSAria SORP.

**Histology and tissue staining**. For observation of histological features, brains were removed, fixed with 4% PFA for 24 h at 4 °C, sectioned at a thickness of 4 μm using an essential microtome (Leica RM2125 RTS), and stained with hematoxylin (DaKo) and 0.25% eosin (Merck). Prior to immunohistochemical and immunofluorescence staining for cancer stem cell markers, macrophage-associate markers and the ARS2-associated factors, $PGE_2$ (Abcam, ab2318, 1:100), Nestin (BD Biosciences, 611658, 1:500), GFAP (MP Biomedicals, 691102, 1:500), Iba-1 (Wako, 019-19741, 1:100), CD86 (EP1158Y, Abcam, ab53004, 1:500), CD206 (Abcam, ab64693, 1:500), Arginase-1 (E-2, Santa Cruz Biotech, sc-271430, 1:500), ARS2 (GeneTex, GTX119872, 1:500), and MAGL (clone 1B1, LSbio, LS-C173047, 1:500), CD44 (146-3C11, Cell Signaling Technology, #3570, 1:500) sections were subjected to an antigen retrieval process using citrate buffer (pH 6.0), and endogenous peroxidase was blocked by incubating with 3% hydrogen peroxide. Tissue sections were then incubated overnight at 4 °C in a humidified chamber with primary antibody, diluted with antibody diluent buffer (IHC World). Tissue sections for DAB staining were developed using 3,3′-diaminobenzidine (DAB; Vector Laboratories) as the chromogen. For immunofluorescent staining, sections stained with primary antibody were subsequently incubated with the appropriate Alexa Fluor 488- or Alexa Fluor 568-conjugated secondary antibody (Thermo-Fisher Scientific).

**Bioinformatics analysis**. SRRT mRNA expression and patient survival plots, grouped by SRRT levels, were derived from the REMBRANDT database of the National Cancer Institute. All data in REMBRANDT, including microarray gene expression data, copy number arrays and clinical phenotype data from glioma specimens, are currently hosted by the Georgetown Database of Cancer (GDOC). All statistical analyses, evaluations of gene expression, and Kaplan–Meier estimations were performed using GraphPad Prism 5 and 7 (GraphPad Prism). Genomic and clinical data for glioma samples were downloaded from the TCGA data portal (http://cancergenome.nih.gov/). TCGA RNA sequencing data were analyzed using BAM files obtained from the Cancer Genomics Hub (http://cghub.ucsc.edu). Expression measurements and RPKM (Reads Per Kilobase of transcript per Million mapped reads) values were estimated using the R package, DEGseq. Also, vehicle or JZL184-treated M2 macrophage RNA sequencing were analyzed using ssGSEA2-2.2.1 (GSEA) for gene set results.

**RNA-sequencing data processing**. RNA-Seq libraries were prepared using the TruSeq RNA Library Prep kit (Illumina) and were sent out for transcriptome resequencing. The Phred quality score of the obtained raw FASTQ files was checked using FastQC (www.bioinformatics.babraham.ac.uk/projects/fastqc). The sequences were mapped onto the hg19 and GRCh37 human genome using Subread aligners (v1.5.3)[48], and each resulting SAM file was analyzed using featureCounts[49] and SeqMonk software (v1.38.2, www.bioinformatics.babraham.ac.uk/projects/seqmonk). For identification of differentially expressed genes (DEGs), read counts generated from featureCounts were normalized and quantified using the LPEseq package[50], which is designed for non-replicated samples. The counts from SeqMonk were generated using the RNA-Seq quantification pipeline of the software. Genes with statistical significance (p-value and q-value with <0.01) and fold-change (±1.5) were chosen as significant DEGs. The resulting graphs of DEGs were represented using the Multiple Experiment Viewer (MeV; v4.9.0)[51] of the TM4 software suite.

Next is RNA-sequencing data processing of subcutaneous mouse model treated with JZL 184. In order to construct cDNA libraries with the TruSeq Stranded mRNA kit, 1ug of total RNA was used. The protocol consists of polyA-selected RNA extraction, RNA fragmentation, random hexamer primed reverse transcription, and 100 nt paired-end sequencing by Illumina HiSeq4000. The libraries were quantified using qPCR according to the qPCR Quantification Protocol Guide and qualified using an Agilent Technologies 2100 Bioanalyzer. We preprocessed the raw reads from the sequencer to remove low quality and adapter sequence before analysis and aligned the processed reads to the *Mus musculus* (mm10) using HISAT v2.0.5[52]. HISAT utilizes two types of indexes for alignment (a global, whole-genome index and tens of thousands of small local indexes). These two types' indexes are constructed using the same BWT (Burrows–Wheeler transform)/a graph FM index (GFM) as Bowtie2. Because of its use of these efficient data structures and algorithms, HISAT generates spliced alignments several times faster than Bowtie and BWA widely used. The reference genome sequence of *Mus musculus* (mm10) and annotation data were downloaded from the UCSC table browser (http://genome.uscs.edu). After alignment, StringTie v1.3.3b was used to assemble aligned reads into transcripts and to estimate their abundance. It provides the relative abundance estimates as FPKM values (Fragments Per Kilobase of exon per Million fragments mapped) of transcript and gene expressed in each sample. FPKM values have already been normalized with respect to library size, so these values are used for comparative analysis of differentially expressed genes between samples.

**Statistics and reproducibility**. All data are expressed as means ± standard error of the mean (SEM) from at least three independent experiments. The Kaplan–Meier method was used to plot survival curves. In the case of patients who were alive at the time of last follow-up, survival records were censored in our analysis. The Statistical Package for the Social Sciences software (version 16; SPSS, Chicago, IL, USA) was used for statistical analysis. In the case of mouse experiments, results of multiple datasets were compared by analysis of variance (ANOVA) using the log-rank (Mantel-Cox) test. The results of two-dataset experiments were compared using a two-tailed Student's $t$-test. $P$-values < 0.05 were considered statistically significant; individual $P$-values are provided in figure legends. Three technical replicates were performed for all experiments for reproducibility.

**Reporting summary**. Further information on research design is available in the Nature Research Reporting Summary linked to this article.

## Data availability

All ARS2 (SRRT) bioinformatic data was collected from REMBRANDT (currently hosted by GDoC), TCGA data portal (http://cancergenome.nih.gov/) and the Cancer Genomics Hub (http://cghub.ucsc.edu). RNA sequencing data have been uploaded in European Genome-phenome Archive (EGA) with EGA-box-1261 accession code and NCBI Gene Expression Omnibus (GEO) with GSE150630 and GSE150631. Accession codes for all datasets will be available without any restriction following Nature policy.

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

## Acknowledgements

We specially appreciate Dr. Myung-Jin Park (Korea Institute of Radiological and Medical Science, Korea) for proving GSCs. We also thank Ms. Mi Ae Kim and Mr. Tae-Sik Kim for excellent technical support in confocal laser-scanner and FACS operation, respectively. We are also grateful for Yoona Seo, Yura Kang, and Woosun Baek for the technical assistance during revision process. This research was supported by grants from the National Cancer Center, Republic of Korea (NCC-1810121, NCC-1810861, NCC-2010273, and NCC-2010320), Basic Science Research Program through the National Research Foundation of Korea (NRF) funded by the Ministry of Science and ICT (NRF-2015R1C1A1A01054963, NRF-2018R1C1B6004768, and NRF-2019R1A2C1089145), Basic Research Lab Program through the National Research Foundation of Korea (NRF) funded by the Ministry of Science (NRF-2018R1A4A1025860), and the Korea Research Fellowship Program through the National Research Foundation of Korea (KRF) funded by Ministry of Science and ICT (NRF-2015H1D3A1036090).

## Author contributions

J.Y., A.I., J.H.K., and J.B.P. designed experiments, analyzed data. J.Y., E.C., S.S.K., Y.T.O., J.H.H., S.H.P., J.H.K., W.L., and Y.Y. performed experiments. T.H.K., X.J., and J.K.S. performed bioinformatics analysis. J.H.K. and J.S. performed FFA analysis. H.K., J.L., D.H.N., K.S.C., and B.Y. provided intellectual support in this study. H.S.G. and H.Y. provided clinical samples and advice. J.Y., E.C., S.S.K., Y.T.O., A.I., J.H.K., and J.B.P. wrote the manuscript.

## Competing interests

The authors declare no competing interests.
