## [Peer Review File · Nature Communications]

Reviewers' Comments:

Reviewer #2:

Remarks to the Author:

Overall Comments: In the submitted manuscript, the authors set out to link the transcription factor ARS2, previously shown to control SOX2 expression and neural stem cell identity and self-renewal, to glioblastoma progression. The proposed link involves transcriptional activation of MAGL causing production of prostaglandin E2 causing M2 tumor associated macrophage polarization to drive a cancer stem cell phenotype. However, the path from ARS2 to macrophage polarization is very convoluted and many of the links are insufficiently established. Without the links between steps, the novelty of the manuscript is greatly diminished, as many independent parts (e.g. PGE2 causing macrophage polarization, macrophage polarization affecting GSCs) have been previously published. Additionally, much of the data shown in the manuscript is presented as difficult to interpret immunofluorescence images rather than more quantitative assays.

Specific Comments:

Major Comments:

1. The manuscript lacks a cohesive narrative. The authors show that ARS2 is slightly elevated in glioma compared to non-tumor (1.5 fold according to figure 1A). They then show that dramatically altering ARS2 expression via overexpression constructs or shRNA has minor effects on MAGL expression. Next, dramatic changes in MAGL expression (overexpression or shRNA) are purported to show changes in PGE2 production, via an immunofluorescence image of less than ten cells. Then the authors switch to directly adding high concentrations of PGE2 to cells. By artificially amplifying the magnitude of changes at each step, the authors are artificially inflating the observed effects – given the actual magnitudes shown, it seems highly unlikely that a 1.5 fold increase in ARS2 expression following tumorigenesis will drive any significant downstream effects that the authors present. This is especially important given that many of the separate findings lack novelty without the connection to the previous findings.
2. A large proportion of the data in the paper is presented as single immunofluorescence images depicting very few cells, and any presented quantification is questionable. Notably, related to figure 6d, the authors state that shARS2 completely eliminates PGE2 and the bargraph in 6e shows 0% PGE2 positive cells, however there is very clearly visible red signal in 6d. The authors should include a quantitative experimental approach for each analysis that currently relies on IF. This is especially problematic given that when quantitative data is shown, the results are quite modest (e.g. protein data in figure 3, the effects of PGE2 in figure 6a).
3. The ARS2 ChIP showing binding at the MGLL promoter is a critical link for ARS2 to the rest of the manuscript, but there are a lot of problems with these data. First and foremost - the labeling is unclear, but it appears that Supplementary Figure 3 presents negative data from a ChIP experiment. It is not explained in the text, but assuming that H3 represents a ChIP for histone H3, then there is no enrichment for any MGLL promoter sequence by ARS2 ChIP versus either input control or the histone H3 ChIP. These are the more accurate controls versus an IgG, which should pull down no DNA. The differences in figure S3 appear to be restricted to primer efficiency – primer 4 seems to amplify the most efficiently based on input and H3. Additionally, the Genetex ARS2 antibody mentioned in the methods is not listed as approved for ChIP, and there are no validation experiments presented by the authors.
4. The authors do not actually test stemness in glioblastoma cells. The limiting dilution sphere formation assay performed in figures 2 and 4 measures a combination of proliferation/survival and self-renewal. Based on the other data presented (figs 2c, 2f especially), it appears as though the knockdowns are simply causing either cell cycle arrest or cell death. In order to actually address stemness, the authors would need to perform serial LDAs to show effects on sustained sphere formation. An analysis of non-stem glioma cells could also be potentially useful.
5. Additional problems with the LDAs – in figure 4h, the control X01 GSC line has a higher stem cell fraction than the MAGL overexpression construct in the same cell line in figure 4j. In figure 4j, the control has a much lower stem cell frequency than the 4h control. Why are the controls so different? Also, the ELDA program used by the authors presents both the graph and an estimate of

the stem cell frequency, these numbers should be presented. Finally, the timescale between plating and counting spheres is not mentioned in the methods.

6. In figure 8a, the authors present an analysis of RNAseq data to argue for an overlap between JZL184 genes and M2 signature genes. The text is slightly unclear, but it seems the RNAseq data is generated from macrophages isolated from subcutaneous xenograft models and then aligned to the human genome (from methods - hg19 specifically). Why wasn't this data aligned to a mouse genome (e.g. mm9)? The infiltrating macrophages should be from the mouse host, rather than GSC-derived? The methods section for the RNAseq is missing essential information, specifically read length and alignment parameters such as mismatch permission, which would affect the ability of mouse RNA to align to the human genome, but this data is suspect and should be re-aligned and reanalyzed.

Other comments:

1. Figure 6b is difficult to interpret, and given that it is an analysis of surface markers on macrophages, should probably be performed using flow cytometry rather than imaging.
2. In figure 6d/f there should be some kind of stain to distinguish human cells from mouse cells when looking at cancer or macrophage markers. The images are too low resolution to distinguish the characteristic nuclear distinctions between species.
3. For the mouse tumor formation studies, median survival times should be presented.
4. Figure 1f lacks a loading control.
5. The methods section is lacking many critical details on experiments.
6. For the genome-wide analysis of differentially expressed genes, a P-value of 0.05 is far too high. Some form of multiple testing correction is essential for such a large analysis.

Reviewer #3:

Remarks to the Author:

Yin et al uncover a tumor-promoting signalling axis, whereby ARS2 upregulates MAGL in glioblastoma stem cells, which leads to an increased production of prostaglandin E2, cancer stemness and M2-like TAM orientation. To my knowledge, this is a novel tumor-promoting pathway with potentially clinical relevance. Important issues remain:

- Fig 3. One siRNA knocks down ARS2 completely but not MAGL, suggesting that part of the MAGL expression is ARS2-independent. Conversely, the other siRNA only knocks down ARS2 partially, but completely abolishes the expression of MAGL. How to explain?
- Fig 5c. beta-catenin as a whole does not seem to be affected. Sometimes there is even an increase in beta-catenin. This contradicts the statement that MAGL knockdown decreased b-catenin accumulation in GSCs.
- The authors treated WT cells with PGE2 to show that b-catenin expression is increased (Fig 5e) and the sphere-forming ability was increased (Sup Fig 5d). These experiments should be done in MAGL KD cells, not in WT cells, to firmly establish a reversal of the KD phenotype by PGE2 administration.
- P11, line 230-231. The authors suggest a link between a decreased Iba1 staining and tumorigenicity of GSCs. There is no clear link provided between these two phenomena.
- All the in vitro work with PGE2 on peritoneal macrophages is of little relevance for TAM. Moreover, the effect of PGE2 has been reported before and conclusions are too simplistic, only based on a selection of a very limited number of genes. One can prove anything if a very limited selection of genes is shown. The in vitro effect of PGE2 on bona fide TAM from ARS2 or MAGL KD tumors would be much more instructive and needs to be checked.
- The authors also need to design experiments to prove that PGE2 is really responsible for instructing the TAM phenotype in vivo. Do TAM really enhance GSC stemness via PGE2 in vivo?
- Line 248. Another disturbing point is that IL-4-polarized peritoneal macrophages are all of a sudden equal to M2-like TAM. I am convinced there are huge differences. In my opinion, there is no relation between both types of macrophages, which are derived from different tissues and have

seen different triggers.

- How does the JZL-148 inhibitor work? Inhibitors usually do not cause the breakdown of its target protein, merely inhibit its function. Why is there a breakdown here?
- Why is the inhibitor tested on subcutaneous and not orthotopic models? Should work as well
- For RNAseq, the authors used CD11b+ cells from tumors. This is absolutely not state-of-the-art! CD11b+ cells include all myeloid cells, which in a tumor include monocytes, neutrophils, eosinophils, mast cells, macrophages, DC. The authors should prove that macrophages are the cells being affected, not the other ones.

<Responses to reviewers' comments>

Reviewer #2 (Remarks to the Author):

Overall Comments: In the submitted manuscript, the authors set out to link the transcription factor ARS2, previously shown to control SOX2 expression and neural stem cell identity and self-renewal, to glioblastoma progression. The proposed link involves transcriptional activation of MAGL causing production of prostaglandin E2 causing M2 tumor associated macrophage polarization to drive a cancer stem cell phenotype. However, the path from ARS2 to macrophage polarization is very convoluted and many of the links are insufficiently established. Without the links between steps, the novelty of the manuscript is greatly diminished, as many independent parts (e.g. PGE2 causing macrophage polarization, macrophage polarization affecting GSCs) have been previously published. Additionally, much of the data shown in the manuscript is presented as difficult to interpret immunofluorescence images rather than more quantitative assays.

In this paper, we demonstrated complicated mechanism explaining communications between GSCs and tumor-associated macrophages (TAMs). These interactions are mediated by comprehensive intercellular signaling that was summarized in unpublished Fig. 1. As the reviewer pointed out, we also concede that there is incompleteness of our explanation. We tried to establish convincing molecular evidence to support our results through this instructive revision process. By looking at our complete and detailed response, we assure that reviewers can discern our improvements and novelty. We hope that our improved supporting data and explanation will be satisfactory and alleviate the reviewer's concern. Since we observed T-cell infiltration signature in gene set enrichment analysis (GSEA) (Original Supplementary Fig. 7e,f), we added a new result for combination therapy of JZL184 and anti-PD-1 monoclonal antibody in syngeneic mouse model. Combination treatment exerted a synergistic effect that the median survival of orthotropic syngeneic mice treated with Vehicle, PD-1, JZL184, or PD-1+JZL184 was 26 days, 29 days, 30.5 days, and 69.5 days, respectively (Revised Fig. 8f). Our new results greatly enhance the novelty and clinical importance of our study.

Unpublished Fig. 1. A schematic model illustrates the signaling and function of ARS2/MAGL on tumor-associated macrophage (TAM) polarization.

Specific Comments:

Major Comments:

1. The manuscript lacks a cohesive narrative. The authors show that ARS2 is slightly elevated in glioma compared to non-tumor (1.5 fold according to figure 1A). They then show that dramatically altering ARS2 expression via overexpression constructs or shRNA has minor effects on MAGL expression. Next, dramatic changes in MAGL expression (overexpression or shRNA) are purported to show changes in PGE₂ production, via an immunofluorescence image of less than ten cells. Then the authors switch to directly adding high concentrations of PGE₂ to cells.

By artificially amplifying the magnitude of changes at each step, the authors are artificially inflating the observed effects – given the actual magnitudes shown, it seems highly unlikely that a 1.5 fold increase in ARS2 expression following tumorigenesis will drive any significant downstream effects that the authors present. This is especially important given that many of the separate findings lack novelty without the connection to the previous findings.

☞ We thank the reviewer for this comment, for we totally agree on the clarifying issue. Although ARS2 mRNA level was 1.5 times higher in glioma than normal tissue (Original Fig. 1a), we thought protein expression might not be exactly correlated. To address this issue, we quantified protein expression from cohort from National Cancer Center Korea (NCC) in original Fig. 1f and Supplementary Fig. 1. ARS2 protein was barely detectable in normal brains, but was widely and strongly expressed in patient tumor samples. These results support that our experimental overexpression of ARS2 is supposed to be within the clinically reproducible and reasonable alteration. Moreover, the knockdown of ARS2 expression clearly reduced MAGL expression (Revised Fig. 3d,e) and the overexpression of MAGL in ARS2-knockdown X01 cells restored GSC self-renewal, which approves the functional association of ARS2-MAGL in regulation of GSC stemness (Original Fig. 4j). Our results strongly support ARS2/MAGL signaling in GSC stemness regulation.

Next, our original immunofluorescence images covered too large extent with many cells to drop the visibility and details when downsizing into a figure format. Therefore, we cropped the representative parts with PGE₂ patterns in MAGL-knockdown GSCs from the original images, which increased the visibility but included lesser number of cells. If the reviewer would suggest uploading the original images, we supplement it in our figures. Moreover, to resolve the reviewer's concern, in our revised figures, we will present carefully quantified information for all of our

immunofluorescence results and keep our higher-visibility images (Revised Supplementary Fig. 5b–e; revised Fig. 6e,g; revised Fig. 7e). We hope that our revised data is satisfactory.

Concentration of PGE₂ treatment is also based on previous scientific references regulating gene expression on PGE₂ administration in macrophages at the level of 10 μM^{1,2}. Based on the previous studies, we treated PGE₂ at 3 different levels up to the previously reported maximum concentration (0.1, 1, and 10 μM). Then, we conducted further experiments with 10μM PGE₂ for we clearly observed full accumulation of β-catenin as shown in immunoblot staining (Original Fig. 5e).

We highly value the reviewer's comment on the cohesive narrative within clinical relevance and significance. As we explained above, we developed our logic based on evidence and clinical observation, and conducted our experiments within reproducible and reasonable setting, rather artificial inflation.

2. A large proportion of the data in the paper is presented as single immunofluorescence images depicting very few cells, and any presented quantification is questionable. Notably, related to figure 6d, the authors state that shARS2 completely eliminates PGE2 and the bargraph in 6e shows 0% PGE2 positive cells, however there is very clearly visible red signal in 6d. The authors should include a quantitative experimental approach for each analysis that currently relies on IF. This is especially problematic given that when quantitative data is shown, the results are quite modest (e.g. protein data in figure 3, the effects of PGE2 in figure 6a).

☞ We thank the reviewer's comments. As we previously explained, we selected a representative part of entire immunofluorescence images to obtaining better visibility and details. Next, we totally apologize for our inadequate quantification data in original Fig. 6d,e. As pointed out by the reviewer, we carefully and repeatedly measured the qualification of positive staining in immunofluorescence images (Revised Fig. 6e). In addition, in our revised figures, we will present accurate quantification data for all of our immunofluorescence results with our higher-visibility images (Revised Supplementary Fig. 5b–e; revised Fig. 6e,g; revised Fig. 7e). If the reviewer would also suggest uploading the original images, we supplement it in our figures. We genuinely hope that our revised quantified data is satisfactory.

3. The ARS2 ChIP showing binding at the MGLL promoter is a critical link for ARS2 to the rest of the manuscript, but there are a lot of problems with these data. First and foremost - the labeling is unclear, but it appears that Supplementary Figure 3 presents negative data from a ChIP experiment. It is not explained in the text, but assuming that H3 represents a ChIP for histone H3, then there is no enrichment for any MGLL promoter sequence by ARS2 ChIP versus either input control or the

histone H3 CHIP. These are the more accurate controls versus an IgG, which should pull down no DNA. The differences in figure S3 appear to be restricted to primer efficiency – primer 4 seems to amplify the most efficiently based on input and H3. Additionally, the Genetex ARS2 antibody mentioned in the methods is not listed as approved for CHIP, and there are no validation experiments presented by the authors.

☞ As the reviewer #2 pointed out, we really apologize for our careless arrangement and description of ChIP data presented in the former manuscript. First, we fixed the labeling of those figures and description in entire text for better clarification. Second, to relieve reviewer's concern for ChIP experiment, we carefully designed two additional oligomer pairs (primer 2 and 3 in revised Supplementary Fig. 3a,b) instead of former primer set 2 (former primer set 3 and 4 correspond to primer set 4 and 5 in revised Fig. 3f,g and Supplementary Fig. 3a,b). In the revised text, the new constructs cover -361~+53 of MAGL promoter region which shows efficient association with histone H3 (positive control) but not endogenous ARS2 (Revised Supplementary Fig. 3a,b). These data clearly demonstrate that the ARS2 is specifically associated with downstream promoter region of MAGL.

As the reviewer #2 has enormous concern for the grade and/or quality of the ARS2 antibody for the ChIP experiment, we have sent the regarding questionnaire to the antibody generating company GeneTex. However, we just received a formal answer from the company.

[Mail attached:]

“Dear Dr. Jong Heon Kim,

This is Jane over GeneTex International Corp. Thank you for contacting us regarding GTX119872. I have confirmed all testing result of GTX119872, unfortunately, it has never tested by ChIP. Based on our company's policy, if antibodies do not test the application, we could not guarantee. Therefore, it is possible this antibody did not test for ChIP in our lab, but you got the expected result in your experiment. We hope this information will be useful. Please let me know if there is anything I can assist.

Best regards,

Jane Chuang

Product Specialist

Marketing & Sales Division

Tel: 886-3-6208988

ext. 6215

Fax: 886-3-6209098”

Therefore, to confirm the quality of the antibody for ChIP, we have tried to verify the process by our own hands based on the former direction³. Efficient antibody for immunoprecipitation should be good for ChIP as well. This is more likely to be true for a polyclonal antibody since the epitope for a monoclonal antibody may block the interaction. The GeneTex ARS2 antibody is

polyclonal antibody, generated by immunizing recombinant protein encompassing a sequence within the N-terminus region of human ARS2 in which DNA association seems not to be interfered.

First, whether the antibody is able to immunoprecipitate endogenous ARS2 or not (this is the most important aspect and prerequisite of antibody quality for CHIP), we examined co-immunoprecipitation with protein-A/G bead (Santa Cruz Biotech) and following western blot with same antibody⁴. As shown in the unpublished Fig. 2a,b, the antibody precipitated endogenous human ARS in both glioma stem cell X01.

To further consolidate data, we run the samples on the 4~12% TGX gel by scaling up the immunoprecipitation reaction and exerted Colloidal Blue staining. As shown in unpublished Fig. 2c (Left image), we could visualize precipitated endogenous ARS2 in the anti-ARS2 lane sharply. The unique and expected band (~100-kDa) of ARS2 was excised from the gel, and then the LC-MS/MS was assessed for the identification of the specifically precipitated band. As shown the in the unpublished Fig. 2c (Right table), the most enriched peptides of the matched protein at this location is undoubtedly ARS2.

More importantly, to confirm the transcriptional relationship between ARS2 and MGLL, we used a lentiviral-based reporter system to monitor the expression of a luciferase reporter gene linked to the upstream promoter region, identified by CHIP, containing binding site(s) for ARS2. As shown in original Fig. 3h, ARS2 knockdown in X01 cells significantly decreased relative luciferase expression, indicating reduced transcriptional activity of ARS2 toward MGLL.

Taken together, all these data support that the antibody is appropriate for the immunoprecipitation and even CHIP experiment. Moreover, the revised data are highly reliable in respect to validate the physical and functional association between the ARS2 and MAGL promoter. We hope that our improved supporting data and explanation will alleviate the reviewer's concern.

Unpublished Fig. 2. Immunoprecipitation of endogenous ARS2 with GTX119872, rabbit polyclonal antibody specific for ARS2/SRRT in X01 glioma stem cells. a, Ponceau S staining result after 4~12%

TGX gel running and blotting into nitrocellulose membrane. b, Western blot result. c, Colloidal Blue staining result. The bands expected to immunoprecipitated ARS2 and corresponding region in control IgG lane were excised and assessed for the liquid chromatography tandem mass spectrometry (LC-MS/MS). Overlapped peptides in the control as compared with anti-ARS2 region were excluded for the counting. H.C. immunoglobulin heavy chain, L.C. immunoglobulin light chain.

4. The authors do not actually test stemness in glioblastoma cells. The limiting dilution sphere formation assay performed in figures 2 and 4 measures a combination of proliferation/survival and self-renewal. Based on the other data presented (figs 2c, 2f especially), it appears as though the knockdowns are simply causing either cell cycle arrest or cell death. In order to actually address stemness, the authors would need to perform serial LDAs to show effects on sustained sphere formation. An analysis of non-stem glioma cells could also be potentially useful.

✍ We thank the reviewer for raising this point. By conducting cell proliferation assay and LDAs, we explained two independent roles of ARS2 knockdown in manipulating proliferation and self-renewal of GSCs. As described in methods and materials, proliferation assay and LDAs are certainly different in their purposes and protocols. Methods of proliferation assay is plating 1×10^3 amount of cells per well in a 96 plate to determine the number of viable cells, based on the ATP present, which is an indicator of metabolically active cells (The CellTiter-Glo Luminescent Cell Viability Assay). General protocol of LDA is plating 100, 50, 25 and 5 cells in a 96 plate to count the number of wells bearing standard spheres of cells to measure sphere forming ability of cancer stem cells.

Therefore, in the aspect of cell proliferation, as the method is described above, we measured the number of viable cells, based on the ATP present. So, the decreased level of relative cell growth in day 5 meant not cell death, but the reduced cell growth and ATP production upon the ARS2 knockdown. Rather, the decreased GSC proliferation might be attributable to decreased self-renewal as shown in fig. 2b,e and additionally conducted serial LDAs on several passages as reviewer suggested (Unpublished Fig. 3).

Furthermore, proliferative activity of normal human immortalized astrocyte was not significantly reduced upon knockdown of ARS2 expression, as shown in unpublished Fig. 4. So, we suggest that ARS2-derived regulation of cell proliferation is specific to cells harboring self-renewal capacity. Taken together, knockdown of ARS2 blocks GSC's self-renewal, which results in decreased proliferation of GSCs.

Unpublished Fig. 3. Serial limiting dilution assays (LDAs) performed with 528 and X01 GSCs. a, LDAs using 528-shCtrl, 528-shARS2 #1, and 528-shARS2 #2 cells at passage 1 (P1), passage 2 (P2) and passage 3 (P3). b, LDAs using X01-shCtrl, X01-shARS2 #1, and X01-shARS2 #2 cells at P1, P2 and P3.

Unpublished Fig. 4. Proliferative activity of normal human immortalized astrocyte. a, Immunoblot (IB) analysis of ARS2 in astrocytes infected with a shARS2-expressing lentiviral or shCtrl construct. b, Cell proliferation assays performed using astrocyte-shCtrl, astrocyte-shARS2 #1, and astrocyte-shARS2 #2 cells.

5. Additional problems with the LDAs – in figure 4h, the control X01 GSC line has a higher stem cell fraction than the MAGL overexpression construct in the same cell line in figure 4j. In figure 4j, the control has a much lower stem cell frequency than the 4h control. Why are the controls so different? Also, the ELDA program used by the authors presents both the graph and an estimate of the stem cell frequency, these numbers should be presented. Finally, the timescale between plating and counting spheres is not mentioned in the methods.

☞ We thank the reviewer's comment on detailed methods and materials. LDA experiment setting conducted for original Fig. 4h,j was identical that the same amount of cells was seeded as described in method section. There are two reasons why stem cell fraction of X01 GSC was differently delineated. First, there was time difference from plating to counting cells with spheres for original Fig. 4h,j which were 16 and 7 days, respectively. As stemness grows stronger over time, we can easily expect that the stemness of the control X01 GSC in original fig. 4h harbors higher stemness.

Second, the purpose of experiments in original Fig. 4h,j is slightly different: the former is to measure self-renewal of GSC (X01)-infected by shMAGL, and the latter is to verify the recovery of GSC self-renewal upon the effect of ARS2-regulated MAGL expression. Therefore, the control X01 GSC used in the recovery experiments was infected twice by control vectors of shARS2 and MAGL-overexpressing viruses, while the control in the original Fig. 4h was infected once by control vector of shMAGL virus. So, lower stem cell fraction in original Fig. 4j could be additionally affected by double vector infection. Therefore, we have added more detailed information on experimental settings including cell numbers and time scales for each figure. Please see our revised Fig. 4h,j legends as attached below.

➤ Page 41 line 4 – line 10

h, LDAs, performed in X01-shCtrl, X01-shMAGL-1, and X01-shMAGL-3 cells. The well-formed sphere was counted at day 16.

j, LDAs, performed in GSCs (X01 cells) infected with a shARS2- or MAGL-expressing lentiviral construct, both shARS2- and MAGL-expressing lentiviral constructs, or a control construct. The well-formed sphere was counted at day 7.

6. In figure 8a, the authors present an analysis of RNAseq data to argue for an overlap between JZL184 genes and M2 signature genes. The text is slightly unclear, but it seems the RNAseq data is generated from macrophages isolated from subcutaneous xenograft models and then aligned to the human genome (from methods - hg19 specifically). Why wasn't this data aligned to a mouse genome (e.g. mm9)? The infiltrating macrophages should be from the mouse host, rather than GSC-derived? The methods section for the RNAseq is missing essential information, specifically read length and alignment parameters such as mismatch permission, which would affect the ability of mouse RNA to align to the human genome, but this data is suspect and should be re-aligned and reanalyzed.

☞ We deeply appreciate the reviewer's comment, and apologize for our inappropriate method description. In our original manuscript, we mistakenly omitted the experimental procedure for the original Fig. 8a, but it was only inserted the RNA sequencing information related to the original Fig. 3a. Therefore, we have revised the corresponding and appropriate method description of the original Fig. 8a as attached below.

➤ Page 29 line 6 – page 30 line 1

Next is RNA-sequencing data processing of subcutaneous mouse model treated with JZL 184. In order to construct cDNA libraries with the TruSeq Stranded mRNA kit, 1 μ g of total RNA was used. The protocol consists of polyA-selected RNA extraction, RNA fragmentation, random hexamer primed reverse transcription and 100nt paired-end sequencing by Illumina HiSeq4000. The libraries were quantified using qPCR according to the qPCR Quantification Protocol Guide and qualified using an Agilent Technologies 2100 Bioanalyzer. We preprocessed the raw reads from the sequencer to remove low quality and adapter sequence before analysis and aligned the processed reads to the *Mus musculus* (mm10) using HISAT v2.0.5⁵². HISAT utilizes two types of indexes for alignment (a global, whole-genome index and tens of thousands of small local indexes). These two types' indexes are constructed using the same BWT (Burrows–Wheeler transform)/ a graph FM index (GFM) as Bowtie2. Because of its use of these efficient data structures and algorithms, HISAT generates spliced alignments several times faster than Bowtie and BWA widely used. The reference genome sequence of *Mus musculus* (mm10) and annotation data were downloaded from the UCSC table browser (<http://genome.ucsc.edu>). After alignment, StringTie v1.3.3b was used to assemble aligned reads into transcripts and to estimate their abundance. It provides the relative abundance estimates as FPKM values (Fragments Per Kilobase of exon per Million fragments mapped) of transcript and gene expressed in each sample. FPKM values have already been normalized with respect to library size, so these values are used for comparative analysis of differentially expressed genes between samples.

Other comments:

1. using flow Figure 6b is difficult to interpret, and given that it is an analysis of surface markers on macrophages, should probably be performed cytometry rather than imaging.

✎ As suggested by the reviewer, we conducted flow cytometry of surface markers on macrophages. Specifically, we first extracted bone marrow-derived macrophages, rather using peritoneal macrophages, to assay the FACS analysis in that the reviewer #3 (Question No.5) pointed out little relevance of PGE₂ on peritoneal macrophages for tumor-associated macrophages (TAM). In revised Fig. 6c, treatment of LPS induced the highest marker expression of M1-like TAM (CD86) up to 19.3%, while administration of IL-4 and 10 μ M PGE₂ did not. Moreover, the FACS analysis with bone marrow-derived macrophages proved that both IL-4 and PGE₂ prominently upregulated the expression of CD206, a marker of M2-like TAM, compared with LPS treatment (Revised Fig. 6b). To verify integrity of macrophagic association, F4/80 (a marker of macrophage) was co-stained with CD86 or CD206 in all of the FACS analysis. Therefore, through this FACS results, we effectively showed the functional similarity between IL-4 and PGE₂ administration on the bone marrow-derived macrophages.

Additionally, we conducted immunocytochemistry of CD206 and ARG1, representative surface markers for M2-like TAM, with bone marrow-derived macrophages treated with LPS, IL-4 and PGE₂ (Revised Supplementary Fig. 6d). In line with the FACS analysis, both IL-4 and PGE₂ were involved in M2-like polarization. Please see revised Supplementary Fig. 6d in our revised manuscript.

Revise Fig. 6. Expression of CD206 (b) and CD86 (c) in bone marrow derived-macrophages (BMDMs) treated with PGE₂, LPS, IL4, or vehicle control by flow cytometry. F4/80 was co-stained for macrophagic population verification.

2. In figure 6d/f there should be some kind of stain to distinguish human cells from mouse cells when looking at cancer or macrophage markers. The images are too low resolution to distinguish the characteristic nuclear distinctions between species.

As suggested by the reviewer, we repeated the co-staining experiment of M1- and M2-like macrophage markers, CD86 and CD206, respectively with a human specific antibody, Nestin (Abcam, Anti-Nestin antibody [196908] human specific reactivity, ab6320). In Unpublished Fig. 5 we detected that CD206/CD86 (Red) staining was differentiated from the expression of Nestin (Green). For the area where CD206/CD86 and Nestin were expressed nearby, we identified different emission peaks of cells by using LSM 780 confocal laser-scanning microscope (Carl Zeiss), which meant different cell species, indicated by yellow arrows (Unpublished Fig. 5). If the reviewer recommends including this as one of the main figures, we would do so.

Unpublished Fig. 5. Co-immunofluorescence images of human Nestin and mouse CD206 (a) or CD86 (b) in brain tissues from X01 orthotopic xenograft mouse model. Scale bar, 50 μ m.

3. For the mouse tumor formation studies, median survival times should be presented.

☞ We appreciate the reviewer's advice. We have added medial survival information for every tumor formation studies conducted in mouse model in our revised relative figs legends, as indicated below.

➤ Page 40 line 4 – 6

Median survival of the orthotopic mice injected with X01-shCtrl, X01-shARS2 #1, or X01-shARS2 #2 was 36 days, 52 days, and 71.5 days, respectively.

➤ Page 41 line 12 – 13

Median survival of the orthotopic mice injected with X01-shCtrl, X01-shMGLL-1, or X01-shMGLL-3 was 36 days, 42 days, and 49 days, respectively.

➤ Page 43 line 1 – 2

Median survival of the mice treated with vehicle, or JZL184 was 36 days, and 40 days, respectively.

4. Figure 1f lacks a loading control.

☞ We thank the reviewer for raising this point for better clarification. The loading control was already published in our previous paper⁵, so it would be replaced by citing the previous paper. In addition to do that, we included the normalized protein ARS2 expression by using the level of loading control, GAPDH, as shown in original Supplementary Fig. 1a. To avoid further confusion for prospective readers, we have revised our figure legends as below.

➤ Page 39 line 8 – 9

f, Immunoblot (IB) analysis of ARS2 in patient tissues from the National Cancer Center, Republic of Korea. GAPDH was used as a loading control⁴⁵.

BRAIN 2015, published Figure 1.

5. The methods section is lacking many critical details on experiments.

As suggested by the reviewer, we did add many critical details of methods and materials as below.

➤ Page 20 line 3 – 9

293T and GL261 cells were maintained in Dulbecco's modified Eagle's medium (DMEM) supplemented with 10% fetal bovine serum (HyClone). Patient-derived GBM stem cells (528, X01, 578, and 0502) were maintained in DMEM/F-12 supplemented with B27 (Invitrogen), EGF (10 ng/ml, R&D Systems), and bFGF (5 ng/ml, R&D Systems). PGE₂, ICG-001, JZL184 was purchased from R&D systems, Peprotech, Tocris Bioscience, respectively. All cells were repeatedly screened for mycoplasma and maintained in culture for less than 6 months after receipt.

➤ Page 21 line 23 – page 22 line 6

mKLF4, sense 5'-GCTCCTCTACAGCCGAGAAT-3' and antisense 5'-AGCACAACTTGCCCATCAG-3', mMrc1 (CD206), sense 5'-TACCTGAGCCACACCTGCT-3', and antisense 5'-GCGCGTTGCCATGGTTTCC-3', mCD86. 5'-CCCGGATGGTGTGTGGCATA-3' and antisense 5'-TCACAAGGAGGAGGCCACA-3', mMcpt1 (MCP1), sense 5'-GCACTTCTTGCCTTCTGG-3' and 5'-AAACCCACCACATCTGTCCT-3', mTNFa sense 5'-TGGCCAACGGCATGGATCTC-3' and antisense 5'-GGGCAGCCTTGCCCTTGAA-3'

➤ Page 22, line 13 – 22

Cell Proliferation Assays

GSCs were plated at 10^3 cells/well densities in 96-well plates containing DMEM/F-12 supplemented with B27, epidermal growth factor (EGF; 10 ng/ml), and basic fibroblast growth factor (bFGF; 5 ng/ml) for in vitro Proliferation assays. The luminescence of viable cells was detected using CellTiter-Glo Luminescent Cell Viability Assay Kit according to the protocol of the manufacture (Promega). CellTiter-Glo Luminescent Cell Viability Assay is a homogenous method of determining the number of viable cells in culture based on quantitation of the ATP present, which signals the presence of metabolically active cells. The luminescence signal was detected by SpectraMax L Microplate Reader (Molecular Device) according to the manufacturer's protocol.

➤ Page 24 line 3 – 10

Primary antibodies against ARS2 (GeneTex), MAGL (Santa Cruz), pLRP6 (Cell Signaling), LRP6 (Cell Signaling Technology), β -catenin (Cell Signaling Technology), Lamin B (Santa Cruz Biotech), Cyclin D1 (Santa Cruz Biotech), c-Myc (Santa Cruz Biotech), Vinculin (Sigma-Aldrich) and α -tubulin (Santa Cruz) were incubated overnight at 4°C. Immunoreactive bands were visualized using peroxidase-labeled affinity purified secondary antibodies (KPL) and the Amersham ECL prime western blotting detection reagent (GE Healthcare). Some of the western blot images were obtained by C-DiGit® Blot Scanner (LI-COR Biosciences).

➤ Page 24 line 11 – 19

Immunoblot assay for MAGL hydrolase activity

GSCs (1×10^6 cells) were plated in 6 cm dishes with GSC culture media. After 24 hours, cells were treated with JZL184 (1 μ M and 2 μ M) or vehicle (DMSO) for 24 hours and harvested for MAGL hydrolase activity measurements. Proteins were extracted with RIPA buffer with complete protease inhibitors (Roche), and 100 μ g proteins were treated with 2 μ M ActivX TAMRA-FP Serine Hydrolase Probes (Thermo Scientific) for 30 min at room temperature (100 μ l total reaction volume). Reactions were quenched with relative volume of standard 5 \times SDS/PAGE loading buffer (reducing), separated by SDS/PAGE, and visualized in-gel with a Typhoon FLA 700 gel scanner (GE).

➤ Page 24 line 20 – page 25 line 5

Chromatin Immunoprecipitation

For each ChIP reaction, $\sim 1 \times 10^6$ X01 cells were crosslinked with 1% formaldehyde for 10 minutes at room temperature, and genomic DNA was fragmented into ~ 100 –300 bp fragments by sonication according to the manufacturer's instructions (MAGnify Chromatin Immunoprecipitation System, Thermo-Fisher Scientific). DNA-bound ARS2 was immunoprecipitated using an ARS2-specific antibody (Genetex). The associated DNA was then purified and analyzed by qRT-PCR to detect specific DNA sequences within the MGLL promoter that were bound in vivo by ARS2 protein. An antibody against IgG (MAGnify Chromatin Immunoprecipitation System, Thermo-Fisher Scientific) was used as a nonspecific control and histone H3 antibody was used as positive ChIP grade control (Abcam).

➤ Page 25 line 6 – 18

Immunocytochemical staining

GSCs (1.5×10^4 cells) were grown on 8-well chambered culture slides (Nunc). After 24 hours, cells were fixed with 2% paraformaldehyde (PFA) and permeabilized by incubating with 0.25% Triton X-100 for 10 minutes at room temperature (RT). After permeabilization, GSCs were immunostained for the cancer stem cell markers, Nestin (BD Biosciences, 1:500) and GFAP (MP Biomedicals, 1:500), as ARS2 (Genetex, 1:500), PGE₂ (Abcam Abcam, 1:100), and β -

Catenin (Cell Signaling Technology, 1:500), CD86 (Abcam, 1:500), CD206 (Abcam, 1:500), ARG1 (Santa Cruz Biotech, 1:500) by incubating overnight at 4°C in a humidified chamber with primary antibody, diluted for the working concentration with antibody diluent buffer (IHC World). Immunoreactive proteins were visualized with the appropriate Alexa Fluor 488- or Alexa Fluor 568-conjugated secondary antibody (Thermo-Fisher Scientific, 1:500). Nuclei were stained with 4',6-diamidino-2-phenylindole (DAPI; Sigma, 1:50000). Fluorescence images were obtained using an LSM 780 confocal laser-scanning microscope (Carl Zeiss).

➤ Page 26 line 3 – 9

For syngeneic orthotopic model, GL261 cells were injected into 5-week-old female C57BL/6 mice. The tumors were extracted, pooled for each experimental group, and mechanically disaggregated using stainless steel operating scissors. The brain of each mouse was harvested and fixed in 4% PFA. JZL184 (30 mg/kg; Tocris Biosciences) was orally administered daily. PD-1 antibody (10 mg/kg; Bioxcell) was administered once a week for 9 weeks through an intraperitoneal injection. Survival was analyzed using GraphPad PRISM software (version 7; GraphPad PRISM, La Jolla, CA, USA).

➤ Page 26 line 20 – page 27 line 1

Isolation and activation of mouse bone marrow-derived macrophages (BMDMs)

Bone marrow cells were collected from female BALB/c nude mice femurs and tibias, and cultured with 20 ng/ml recombinant M-CSF (BioLegend) for 5 days in a petri dish. On day 5, naïve BMDMs were stimulated with LPS (10 ng/mL; Sigma-Aldrich) or IL-4 (10 ng/mL; Selleckem) to generate the BMDM-M1 or BMDM-M2 macrophages, respectively, for 24h.

➤ Page 27 line 2 – 9

Flow cytometry

For surface marker analysis, live cells were re-suspended in 0.1% BSA 1xPBS and stained with anti-mouse F4/80 (PE, 1:100, Invitrogen), CD11b (PerCP-Cyanine5.5, 1:100 Invitrogen), and CD86 (APC, 1:100, BioLegend) at 4°C for 20 min. Cells were fixed and permeabilized (Cytotfix/cytoperm, BD) for intracellular protein staining, then labeled with anti-mouse CD206 (FITC, 1:100, BioLegend). Data were acquired by the BD LSRFortessa and analyzed with FACS Diva and FlowJo software. Cell sorting was performed by BD Cell Sorting FACSaria SORP.

➤ Page 27 line 15 – 19

PGE₂ (Abcam, 1:100), Nestin (BD Biosciences, 1:500), GFAP (MP Biomedicals, 1:500), Iba-1 (Wako, 1:100), CD86 (Abcam, 1:500), CD206 (Santa Cruz Biotech, 1:500), Arginase-1 (Santa Cruz Biotech, 1:500), ARS2 (GeneTex, 1:500), and MAGL (Lsbio, 1:500), CD44 (Cell Signaling Technology, 1:500) sections were subjected to an antigen retrieval process using citrate buffer (pH 6.0)

➤ Page 29 line 3 – 4

Genes with statistical significance (*p*-value and *q*-value with less than 0.01) and fold-change (± 1.5) were chosen as significant DEGs. The resulting graphs of DEGs were represented using the Multiple Experiment Viewer (MeV; v4.9.0)⁵¹ of the TM4 software suite.

➤ Page 29 line 6 – page 30 line 1

Next is RNA-sequencing data processing of subcutaneous mouse model treated with JZL 184. In order to construct cDNA libraries with the TruSeq Stranded mRNA kit, 1ug of total RNA was used. The protocol consists of polyA-selected RNA extraction, RNA fragmentation, random

hexamer primed reverse transcription and 100nt paired-end sequencing by Illumina HiSeq4000. The libraries were quantified using qPCR according to the qPCR Quantification Protocol Guide and qualified using an Agilent Technologies 2100 Bioanalyzer. We preprocessed the raw reads from the sequencer to remove low quality and adapter sequence before analysis and aligned the processed reads to the *Mus musculus* (mm10) using HISAT v2.0.5⁵². HISAT utilizes two types of indexes for alignment (a global, whole-genome index and tens of thousands of small local indexes). These two types' indexes are constructed using the same BWT (Burrows–Wheeler transform)/ a graph FM index (GFM) as Bowtie2. Because of its use of these efficient data structures and algorithms, HISAT generates spliced alignments several times faster than Bowtie and BWA widely used. The reference genome sequence of *Mus musculus* (mm10) and annotation data were downloaded from the UCSC table browser (<http://genome.uscs.edu>). After alignment, StringTie v1.3.3b was used to assemble aligned reads into transcripts and to estimate their abundance. It provides the relative abundance estimates as FPKM values (Fragments Per Kilobase of exon per Million fragments mapped) of transcript and gene expressed in each sample. FPKM values have already been normalized with respect to library size, so these values are used for comparative analysis of differentially expressed genes between samples.

6. For the genome-wide analysis of differentially expressed genes, a P-value of 0.05 is far too high. Some form of multiple testing correction is essential for such a large analysis.

We thank the reviewer's comment about statistical rationale to select differentially expressed genes (DEGs). As indicated by the reviewer, we have chosen the following DEGs based on three parameters using SeqMonk software and LPEseq package: (1) statistical significance of *p*-value (< 0.01), (2) *q*-value, and (3) fold change with more than 1.5 or less than 1.5. Corresponding information was added and modified in our manuscript.

Significant Genes	P-value	q-value	Fold-Change
LIN28B	0.014	0.16	5.37
HSPA8	0.018	0.19	2.92
EEF2	0.007	0.11	1.57
MYCN	0.007	0.11	1.52
HMGA1	0.02	0.21	1.51
AL593851.1	7.73E-05	0.009	1.38
ACTB	0.0001	0.01	0.49
SH3BGRL	0.019077	0.19	0.27
OAS3	0.031805	0.22	0.27
C20orf30	0.004974	0.088	0.26
MGLL	4.78477E-06	0.001	0.25
VASP	0.031805	0.22	0.24

➤ Page 29 line 3 – 4

Genes with statistical significance (*p*-value and *q*-value with less than 0.01) and fold-change (\pm 1.5) were chosen as significant DEGs.

Reviewer #3 (Remarks to the Author):

Yin et al uncover a tumor-promoting signaling axis, whereby ARS2 upregulates MAGL in glioblastoma stem cells, which leads to an increased production of prostaglandin E2, cancer stemness and M2-like TAM orientation. To my knowledge, this is a novel tumor-promoting pathway with potentially clinical relevance. Important issues remain:

1. Fig 3. One siRNA knocks down ARS2 completely but not MAGL, suggesting that part of the MAGL expression is ARS2-independent. Conversely, the other siRNA only knocks down ARS2 partially, but completely abolishes the expression of MAGL. How to explain?

☞ We really apologize the careless presentation of the data in the former Fig. 3d,e. To resolute the reviewer's concern, we have carefully assessed the knock down experiments again with same shRNAs for the ARS2 to resolve inconsistency of the expression level between ARS2 and MAGL proteins (Revised Fig. 3d,e) For better clearance upon the expression correlation between ARS2 and MAGL, we designed two additional shRNAs targeting ARS2 specifically. As shown in unpublished Fig. 6, MAGL expression is completely correlated with knock down efficacy of ARS2 itself. We hope that our improved data will alleviate the reviewer's concern. Please see Fig. 3d,e of our revised manuscript.

Unpublished Fig. 6. Immunoblot analysis of ARS2 in X01 infected with a new shARS2-expressing lentiviral (#3, #4) or shCtrl construct.

2. Fig 5c. beta-catenin as a whole does not seem to be affected. Sometimes there is even an increase in beta-catenin. This contradicts the statement that MAGL knockdown decreased b-catenin accumulation in GSCs.

☞ We highly appreciate the reviewer's critical comment. We deeply apologize for this contracting result presentation. Incomplete and inefficient segregation of nuclear and cytosol fraction is thought to be the cause of the implausibly high expression of β -catenin in cytosol fraction in that basal β -catenin in nature is highly expressed in nuclear rather than cytosol. Therefore, we repeatedly sorted and fractionated the nuclear and cytosol proteins, and confirmed successful fractionation by Lamin B

expression, which presents in nuclear alone. Then, we measured β -catenin level in MAGL or ARS2 knockdown GSCs 528, clearly demonstrating the decreased β -catenin expression in nuclear fraction (Revised Fig. 5c). Please revise Fig. 5c of our revised manuscript.

3. The authors treated WT cells with PGE₂ to show that b-catenin expression is increased (Fig 5e) and the sphere-forming ability was increased (Sup Fig 5d). These experiments should be done in MAGL KD cells, not in WT cells, to firmly establish a reversal of the KD phenotype by PGE₂ administration.

☞ We thank the reviewer for this point. First of all, GSC 578 cells, utilized to measure β -catenin and LRP6 phosphorylation by PGE₂ treatment, naturally harbor low levels of both ARS2 and MAGL. Unlike the reviewer's concern, 578 cells are not representative of WT cells, rather it might imitate ARS2/MAGL knockdown cells.

As indicated by the reviewer, we additionally conducted the identical experiments by using MAGL-knockdown GSCs X01. Clear dose-dependent relationship between PGE₂ treatment and upregulation of β -catenin and LRP phosphorylation was shown in unpublished Fig. 7a. Further, treatment of PGE₂ stimulated sphere-forming ability in MAGL-knockdown X01 cells (Unpublished Fig. 7b).

Unpublished Fig. 7. MAGL modules GSC self-renewal by regulating PGE₂. a, IB analysis of β -catenin in MAGL-knockdown X01 cells treated with different concentrations of PGE₂ (0.1, 1, 10 μ M). α -tubulin was used as a loading control. b, Sphere-formation assays performed using MAGL-knockdown X01 cells treated with PGE₂, or vehicle. Scale bar, 100 μ m.

4. P11, line 230-231. The authors suggest a link between a decreased Iba1 staining and tumorigenicity of GSCs. There is no clear link provided between these two phenomena.

☞ We thank the reviewer for raising this point. Iba-1 is a widely utilized cell surface marker of tumor associated macrophages (TAMs). The involvement of TAMs referred by Iba-1 staining in regulation of GSC tumorigenicity has been studied in previous studies^{6,7}. Despite lack of clear-cut mechanisms directing Iba-1 at GSC tumorigenicity, Zhou et al. suggested that the enrichment of GSC marker was significantly correlated with TAM density in patient-derived primary GBM tumor samples. The upregulated tumorigenicity of GSCs through recruitment of TAMs is mainly affected by maintaining the pro-tumorigenic M2 TAM, and the correlation between tumorigenicity of GSCs and total TAM density is identifiable. In that regard, our decreased Iba-1 staining followed by ARS2/MAGL knockdown is partially suggestive of decreased tumorigenicity of GSCs, which was confirmed by transition to the M1 TAM polarization. But we accept the reviewer's point about absence of clear link, so we have modified our manuscript to avoid further misunderstanding.

➤ Page 11 line 16 – 19

In these xenograft mouse models, ARS2 or MAGL knockdown decreased staining for Iba-1, a marker of TAMs^{28,29}, suggesting inflammatory signaling associated with ARS2 or MAGL potentially regulates the tumorigenicity of GSCs (Supplementary Fig. 6a,b).

5. All the *in vitro* work with PGE2 on peritoneal macrophages is of little relevance for TAM. Moreover, the effect of PGE2 has been reported before and conclusions are too simplistic, only based on a selection of a very limited number of genes. One can prove anything if a very limited selection of genes is shown. The *in vitro* effect of PGE2 on bona fide TAM from ARS2 or MAGL KD tumors would be much more instructive and needs to be checked.

☞ We greatly appreciate the reviewer's comment. As suggested by the reviewer, we first extracted bone marrow-derived macrophages (BMDM) from mouse tibia and femur, which are permeable into the brain through blood-brain barrier and more relevant to tumor-associated macrophages (TAMs)^{8,9}. Furthermore, BMDM present an ideal *in vitro* model to understand the mechanisms controlling polarization of activated macrophages¹⁰.

The reviewer also concerned about the limited number of genes for TAM polarization. First of all, LPS and IL-4 are the most potent and routinely utilized cytokines to induce phenotype polarization of BMDM into M1- or M2-like TAMs, respectively¹¹⁻¹⁴. Upon the cytokine induction, the polarization of TAM is mainly validated by surface marker expression. However, the first issue is the numerous diversities of the markers, and the second issue is lack of absolute criteria for selection of the markers. Base on that M2-like TAM (alternatively activated macrophage) is activated by IL4 or

IL13 and expresses arginase-1, mannose receptor (CD206) and IL4 receptor- α ¹³. Therefore, to ensure our accurate TAM polarization, we first selected frequently adopted TAM phenotype markers^{7, 12, 15-21}, and then tried to confirm our results through various experimental methods, including real-time PCR, FACS, and immunofluorescence. All of our experimental results constantly proved that the specific TAM polarization upon the PGE₂ treatment. As suggested by the reviewer, we established BMDM along with our previous peritoneal macrophages, and both models also reassured the efficiency of marker expression depending on the TAM phenotype.

Then, we assayed flow cytometry to demonstrate whether *in vitro* effect of PGE₂ treatment on M2-like TAM polarization on the BMDMs (Revised Fig. 6b,c). The treatment of LPS upregulated the highest expression of CD86 (a marker of M1 TAM), along with positive F4/80 staining (a marker of macrophage), up to 19.3%, while administration of IL-4 and 10 μ M PGE₂ did not. Moreover, the FACS analysis proved that both IL-4 and PGE₂ even prominently upregulated the expression of double positivity of CD206 (a marker of M2 TAM) and F4/80, compared with LPS treatment (Revised Fig. 6b). Therefore, through this FACS results, we effectively showed the functional similarity between IL-4 and PGE₂ administration on the BMDMs.

We additionally conducted immunocytochemistry of CD206 and ARG1, representative surface markers for M2-like macrophage with bone marrow-derived macrophages treated with LPS, IL-4 and PGE₂ (Revised Supplementary Fig. 6d). In line with the FACS analysis, both IL-4 and PGE₂ were involved in M2-like TAM polarization.

6. The authors also need to design experiments to prove that PGE2 is really responsible for instructing the TAM phenotype *in vivo*. Do TAM really enhance GSC stemness via PGE2 *in vivo*?

☞ We greatly appreciate the reviewer's comment. Actually, *in vivo* test of PGE₂-educated macrophages on GSC stemness is difficult. In our two independent xenograft mouse models, GSCs stemness referred by altered expression of PGE₂ upon ARS2 or MAGL knockdown was demonstrated (Original Fig. 6d,f). Using an immunofluorescence approach, we showed that decreased expression of PGE₂ shared the downregulation of Nestin, a stemness marker of GSCs, as well as CD206 and ARG1, surface markers of M2-like TAMs. Furthermore, orthotopically implanted GSC X01 cells into the brains of nude mice, orally treated with a MAGL-specific inhibitor (JZL184) showed that JZL184 decreased PGE₂, Nestin, and M2-like TAM expression markers likewise (Original Figure. 7d). Macrophages after PGE₂ treatment also secreted cytokines possibly mediated to upregulation of GSC stemness, including Lipocalin 2, Serpin E1, G-SCF, HGF, VEGF, and IL6 (Original Supplementary Fig. 6d,e). Taken together, our data suggest that the regulation of GSC self-renewal and tumorigenicity through TAM polarization is closely related to the production of PGE₂.

7. Line 248. Another disturbing point is that IL-4-polarized peritoneal macrophages are all of a sudden equal to M2-like TAM. I am convinced there are huge differences. In my opinion, there is no relation between both types of macrophages, which are derived from different tissues and have seen different triggers.

☞ We greatly appreciate the reviewer's advice. As suggested by the reviewer, different tissue-dependent residency of macrophages might have little thing in common. Therefore, we tried to extract bone marrow-derived macrophages (BMDMs) from mouse tibia and femur, rather than using peritoneal macrophages. According to the previous evidence, BMDMs were shown to be permeable into the brain through blood-brain barrier and more relevant to tumor-associated macrophages (TAMs)^{8, 9}; furthermore, BMDM present an ideal *in vitro* model to understand the mechanisms controlling polarization of activated macrophages¹⁰. Then, we further conducted the related experiments with these BMDMs to examine PGE₂-induced macrophage polarization.

The prototypic cytokine IL-4 is known to induce the BMDM polarization into M2 phenotype¹²⁻¹⁴. IL-4 treatment has been rigorously studied as a facilitator of M2-like TAM polarization in various malignant cell lines²²⁻²⁶. Moreover, upon the activation of M2 polarization, IL-4 signaling mediator was shown to be highly expressed in TAMs²³. Taken together, we support that the IL-4-polarized BMDMs share similarity to M2-like polarized TAM.

8. How does the JZL-148 inhibitor work? Inhibitors usually do not cause the breakdown of its target protein, merely inhibit its function. Why is there a breakdown here?

☞ We truly appreciate the reviewer's comment. There was a reference demonstrating the efficacy of JZL184 on regulation of protein MAGL expression²⁷, as we previously included in our results. More specifically, however, JZL184 is supposed to be tested by regulated enzymatic activity of MAGL for MAGL is a serine hydrolase, as indicated by the reviewer. Therefore, we additionally adopted a Thermo Scientific Serine Hydrolase Probes to determine active enzyme levels depending on the JZL184 treatment on MAGL. These probes are ActivX Fluorophosphonate (FP), enabling selective enrichment on functionally active and biologically relevant enzymes, not on inactive proenzymes (Zymogens) in serine hydrolase family. Upon the treatment of JZL184 in the GSC X01 and 528 cells, we clearly observed the decreased enzymatic activity of MAGL as shown in Revised Supplementary Fig. 7a.

Revised Supplementary Fig. 7a. Immunoblot (IB) analysis of MAGL hydrolase activity in GSCs (X01, 528 cells) treated with JZL184 (0, 1, 2 μ M).

9. Why is the inhibitor tested on subcutaneous and not orthotopic models? Should work as well

☞ We already tested the efficacy of MAGL inhibitor, JZL184, in the brain orthotopic model as shown in original Fig. 7. Please see our original Fig. 7 and corresponding description in the manuscript, as indicated below.

➤ Page 13 line 24 – page 14 line 15

Next, we orthotopically implanted GSC X01 cells into the brains of nude mice, and then treated mice orally with JZL184 or vehicle every day. Treatment with JZL184 decreased tumor mass; immunohistochemical staining further showed that JZL184 decreased MAGL levels and the number of Iba-1-expressing cells (Fig. 7c). Notably, administration of JZL184 was sufficient to suppress MAGL expression in GSCs and infiltration of TAMs (Fig. 7c).

Production of PGE₂, the final manifestation of MAGL-induced neuroinflammation, was also extinguished by treatment with JZL184, as demonstrated by immunofluorescence (Fig. 7d). M1-like TAM polarization, represented by the marker CD86, was increased in X01 cells by treatment with JZL184, whereas M2-like TAMs, marked by CD206 and ARG1 expression, exhibited an opposite response to JZL184 administration (Fig. 7d,e). We confirmed that treatment with JZL184 downregulated the stemness marker Nestin, and upregulated GFAP (Fig. 7d,e). Finally, we assessed the survival of xenograft mice following JZL184 treatment. These experiments revealed that survival was significantly longer for JZL184-treated xenograft mice compared with vehicle-treated controls (Fig. 7f). These beneficial effects of JZL184 treatment indicate that pharmacological inhibition of MAGL suppresses the self-renewal and tumorigenic capacity of GSCs and promotes M1-like polarization of TAMs.

10. For RNAseq, the authors used CD11b⁺ cells from tumors. This is absolutely not state-of-the-art! CD11b⁺ cells include all myeloid cells, which in a tumor include monocytes, neutrophils, eosinophils, mast cells, macrophages, DC. The authors should prove that macrophages are the cells being affected, not the other ones.

We highly appreciate the reviewer's comment. As mentioned by the reviewer, CD11b is a marker of macrophages along with other myeloid cells. Rather, F4/80 is a well-known major antibody to stain murine macrophage population²⁸. Because we previously found that the sorted pellets only by CD11b were relatively small, we thought co-stained cells with CD11b and F4/80 would be fewer and inadequate for experimental analysis. This was the reason why we sorted macrophages by a single marker, CD11b, even though the other myeloid cells would be sorted together.

As the reviewer suggested confirming that macrophages were the cells being affected, we re-established subcutaneous mouse models injected with GSC X01, and conducted FACS analysis by CD11b and F4/80 (Unpublished Fig. 8a,b). The macrophage fraction among cells with CD11b+ population was small as 3.4%. However, in the point of F4/80 positively sorted cells, almost all macrophagic population was co-stained with CD11b+, meaning that all macrophages were obviously included in CD11b+ cells. Moreover, absolute immune cell fraction in our GSEA analysis from CD11b+ cells also demonstrated the majority composition was macrophage in both control and JZL184-treated X01 human cells, with 9.56% and 10.77%, respectively (Unpublished Fig. 8c). Effective regulation of M1 and M2 macrophagic gene signature followed by JZL184 treatment further confirmed that the macrophages sorted by CD11b marker were main factor involved in our analysis (Unpublished Fig. 8d). Taken together, our data is strongly representative of regulation of macrophages, and indicative of functional macrophagic association.

Unpublished Fig. 8.

References

1. Na, Y.R., Jung, D., Yoon, B.R., Lee, W.W. & Seok, S.H. Endogenous prostaglandin E2 potentiates anti-inflammatory phenotype of macrophage through the CREB-C/EBP-beta cascade. *Eur. J. Immunol.* **45**, 2661-2671 (2015).
2. Sanin, D.E. *et al.* Mitochondrial Membrane Potential Regulates Nuclear Gene Expression in Macrophages Exposed to Prostaglandin E2. *Immunity* **49**, 1021-1033.e1026 (2018).
3. Wardle, F.C. & Tan, H. A CHIP on the shoulder? Chromatin immunoprecipitation and validation strategies for CHIP antibodies. *F1000Res.* **4**, 235 (2015).
4. Kim, J.H. & Richter, J.D. Opposing polymerase-deadenylase activities regulate cytoplasmic polyadenylation. *Mol. Cell* **24**, 173-183 (2006).
5. Yin, J. *et al.* DEAD-box RNA helicase DDX23 modulates glioma malignancy via elevating miR-21 biogenesis. *Brain* **138**, 2553-2570 (2015).
6. Zhou, W. *et al.* Periostin secreted by glioblastoma stem cells recruits M2 tumour-associated macrophages and promotes malignant growth. *Nat. Cell Biol.* **17**, 170-182 (2015).
7. Shi, Y. *et al.* Tumour-associated macrophages secrete pleiotrophin to promote PTPRZ1 signalling in glioblastoma stem cells for tumour growth. *Nat. Commun.* **8**, 15080 (2017).
8. Takata, K. *et al.* Induced-Pluripotent-Stem-Cell-Derived Primitive Macrophages Provide a Platform for Modeling Tissue-Resident Macrophage Differentiation and Function. *Immunity* **47**, 183-198.e186 (2017).
9. Bowman, R.L. *et al.* Macrophage Ontogeny Underlies Differences in Tumor-Specific Education in Brain Malignancies. *Cell Rep.* **17**, 2445-2459 (2016).
10. Zhuang, G. *et al.* A novel regulator of macrophage activation: miR-223 in obesity-associated adipose tissue inflammation. *Circulation* **125**, 2892-2903 (2012).
11. Krausgruber, T. *et al.* IRF5 promotes inflammatory macrophage polarization and TH1-TH17 responses. *Nat. Immunol* **12**, 231-238 (2011).
12. Biswas, S.K. & Mantovani, A. Macrophage plasticity and interaction with lymphocyte subsets: cancer as a paradigm. *Nat. Immunol.* **11**, 889-896 (2010).
13. Mosser, D.M. & Edwards, J.P. Exploring the full spectrum of macrophage activation. *Nat. Rev. Immunol.* **8**, 958-969 (2008).

14. Qian, B.Z. & Pollard, J.W. Macrophage diversity enhances tumor progression and metastasis. *Cell* **141**, 39-51 (2010).
15. Huber-Ruano, I. *et al.* An antisense oligonucleotide targeting TGF-beta2 inhibits lung metastasis and induces CD86 expression in tumor-associated macrophages. *Ann. Oncol.* **28**, 2278-2285 (2017).
16. Olsson, A. *et al.* Tasquinimod triggers an early change in the polarization of tumor associated macrophages in the tumor microenvironment. *J. Immunother. Cancer* **3**, 53 (2015).
17. Achyut, B.R. *et al.* Canonical NFkappaB signaling in myeloid cells is required for the glioblastoma growth. *Sci. Rep.* **7**, 13754 (2017).
18. Mantovani, A. & Allavena, P. The interaction of anticancer therapies with tumor-associated macrophages. *J. Exp. Med.* **212**, 435-445 (2015).
19. Liu, L. *et al.* Tumor associated macrophage-targeted microRNA delivery with dual-responsive polypeptide nanovectors for anti-cancer therapy. *Biomaterials* **134**, 166-179 (2017).
20. Ge, H. *et al.* Tumor associated CD70 expression is involved in promoting tumor migration and macrophage infiltration in GBM. *Int. J. Cancer* **141**, 1434-1444 (2017).
21. Mota, J.M. *et al.* Post-Sepsis State Induces Tumor-Associated Macrophage Accumulation through CXCR4/CXCL12 and Favors Tumor Progression in Mice. *Cancer Immunol Res.* **4**, 312-322 (2016).
22. DeNardo, D.G. *et al.* CD4(+) T cells regulate pulmonary metastasis of mammary carcinomas by enhancing protumor properties of macrophages. *Cancer Cell* **16**, 91-102 (2009).
23. Pello, O.M. *et al.* Role of c-MYC in alternative activation of human macrophages and tumor-associated macrophage biology. *Blood* **119**, 411-421 (2012).
24. Gocheva, V. *et al.* IL-4 induces cathepsin protease activity in tumor-associated macrophages to promote cancer growth and invasion. *Genes Dev.* **24**, 241-255 (2010).
25. Liu, C.Y. *et al.* M2-polarized tumor-associated macrophages promoted epithelial-mesenchymal transition in pancreatic cancer cells, partially through TLR4/IL-10 signaling pathway. *Lab Invest.* **93**, 844-854 (2013).
26. Wang, H.W. & Joyce, J.A. Alternative activation of tumor-associated macrophages by IL-4: priming for protumoral functions. *Cell Cycle* **9**, 4824-4835 (2010).

27. Ma, M. *et al.* Monoacylglycerol lipase inhibitor JZL184 regulates apoptosis and migration of colorectal cancer cells. *Mol. Med. Rep.* **13**, 2850-2856 (2016).
28. Austyn, J.M. & Gordon, S. F4/80, a monoclonal antibody directed specifically against the mouse macrophage. *Eur. J. Immunol.* **11**, 805-815 (1981).

Reviewers' Comments:

Reviewer #2:

Remarks to the Author:

Overall Comments: In the revised manuscript, the authors attempt to clarify their report, in which the transcription factor ARS2, previously shown to control SOX2 expression and neural stem cell identity and self-renewal, is linked to glioblastoma progression through transcriptional activation of MAGL causing production of prostaglandin E2 causing M2 tumor associated macrophage polarization driving a cancer stem cell phenotype. There were significant problems precluding publication of the initial submission, and the authors have made insufficient progress in addressing these flaws.

Specific Comments:

1. During the last revision period, a paper entitled "Ars2 promotes cell proliferation and tumorigenicity in glioblastoma through regulating miR-6798-3p" (PMID: 30349053) was published.
2. The story remains very convoluted, and the links between the steps, which are critical for the novelty of the paper, are still weak. For instance, the authors contend that the end product of ARS2 upregulation is signaling to cause M2 macrophage polarization, yet, the majority of the results in the paper take place in cell culture with no macrophages present, and the authors purport to show changes in stemness of glioma anyway. GSCs have been reported to cause M2 polarization (PMID: 25580734) so I'm unclear why this inclusion is necessary. Is ARS2 necessary for stemness, or for stem cells to induce M2 polarization, or just for survival of cells in general?
3. The authors have not addressed the concern of artificial amplification of signal at each step.

Specifically:

- a. "Although ARS2 mRNA level was 1.5 times higher in glioma than normal tissue (Original Fig. 1a), we thought protein expression might not be exactly correlated. To address this issue, we quantified protein expression from cohort from National Cancer Center Korea (NCC) in original Fig. 1f and Supplementary Fig. 1. ARS2 protein was barely detectable in normal brains, but was widely and strongly expressed in patient tumor samples. These results support that our experimental overexpression of ARS2 is supposed to be within the clinically reproducible and reasonable alteration." I'm confused by this point – the authors contend that the mRNA is not predictive of protein expression so figure 1A is either incorrect or misleading. Then why are the data still figure 1A? Also, the new data are slightly suspect – the authors suggest ARS2 expression in brain is undetectable, yet ARS2 is a common essential gene, with ARS2 disruption almost universally lethal (<https://depmap.org/portal/gene/SRRT?tab=overview>), including in CNS samples.
 - b. "Concentration of PGE2 treatment is also based on previous scientific references regulating gene expression on PGE2 administration in macrophages at the level of 10 μ M^{1,2}." I don't see the relevance of the references. PGE2 should be added at concentrations that the glioma cells are capable of producing – levels which are never directly measured in the manuscript. What if the positive staining in figures 5a/b reflect nanomolar concentrations of PGE2?
4. The overreliance on immunofluorescence images is still unacceptable. The standard joke in science is that "representative" images means the best image you can find; without a more holistic analysis the data are suspect. It is also unacceptable to force the reader to connect to a database with all of the higher visibility images – the manuscript should be a self-contained package that allows the reader to judge the work. Figures 6d-g are still problematic – even ignoring the doubts about the accuracy of the quantification raised by the previously incorrect results, the PGE2 staining in figure 6d is definitely more red than figure 6f, yet the bar chart in 6g has a higher percent positive than 6e – showing the limitation of representative images.
 5. The authors response to problems with the LDAs are insufficient. LDAs are not designed to measure proliferation, but proliferation is a confounding factor in an LDA – anything that affects proliferation will affect the results of an LDA. Figures 2c and 2f show that proliferation is greatly reduced by shARS2, so even if there is no effect on stemness, the LDA will show less sphere formation. This is where serial LDAs are critical – and a serial LDA involves taking the spheres formed by the initial LDA and replating them, not plating independent serial tissue culture passages for LDAs. Additionally, the time until reading should be consistent between experiments

and 7 days is too short for adequate sphere formation.

a. "As stemness grows stronger over time, we can easily expect that the stemness of the control X01 GSC in original fig. 4h harbors higher stemness." This is an absolutely incorrect statement. Sphere formation increases with time, because of the effects of proliferation on the LDA, but those cells were always stem cells and the wells were false negatives until they were able to grow into a sphere. That's why it's critical to wait until saturation of sphere formation to get a readout of stem cell frequency independent of proliferation effects.

6. The rebuttal states that β -catenin is highly expressed in the nucleus rather than the cytosol, yet the staining in figure 5a/b is clearly cytoplasmic.

7. "Genes with statistical significance (p-value and q-value with less than 0.01) and fold-change (± 1.5) were chosen as significant DEGs." All of the samples on the table except AL593851.1 and MGLL have a q-value above 0.01? Also, was ARS2 itself explicitly excluded, or does it not show up as significantly altered?

Reviewer #3:

Remarks to the Author:

The authors already did an effort to accommodate most of my concerns, but there are still a few important problems:

1) The authors changed peritoneal macrophages into bone marrow-derived macrophages for their in vitro experiments, which is already some improvement, but BMDM are still quite distinct from real TAM. As already asked before, it would be much more relevant and informative to sort real TAM (CD11b+ F4/80+) from ARS2 or MAGL KD tumors and treat those cells with PGE2.

2) The authors could use inhibitors that block PGE2 production in vivo, to test the in vivo relevance of this pathway in their models.

3) RNAseq to screen for macrophage genes was performed on cell preparations that contained less than 10% macrophages. This is unacceptable. How can the authors be sure that the regulated genes are intrinsic to macrophages? At the very least, genes whose expression were altered should be tested on purified TAM populations.

Reviewers' comments:

Reviewer #2 (Remarks to the Author):

Overall Comments: In the revised manuscript, the authors attempt to clarify their report, in which the transcription factor ARS2, previously shown to control SOX2 expression and neural stem cell identity and self-renewal, is linked to glioblastoma progression through transcriptional activation of MAGL causing production of prostaglandin E2 causing M2 tumor associated macrophage polarization driving a cancer stem cell phenotype. There were significant problems precluding publication of the initial submission, and the authors have made insufficient progress in addressing these flaws.

Specific Comments:

1. During the last revision period, a paper entitled “Ars2 promotes cell proliferation and tumorigenicity in glioblastoma through regulating miR-6798-3p” (PMID: 30349053) was published.
 - Response: First, we have noted that the paper mentioned by the referee (PMID: 30349053) was published in the journal Scientific Report on Oct 22, 2018 in the middle of our first-round review process. In first round reviewer’s comment, there were no comments related to this paper, so we thought possibly it’s because the contents of the published manuscript are fundamentally different from the works of our study. The main point of the published paper is elucidating role of ARS2 on target miRNA-793-3p exclusively in differentiated glioblastoma cell lines, which is already well-known function of ARS2. In contrast, we identify the ARS2-associated signaling mechanisms to drive self-renewal and tumorigenicity of GSCs, concurrently with M2-like TAM polarization. Also, the novel mechanisms in our study engage an original role as a transcription factor of ARS2 in GSCs and a number of previously unknown effectors communicating tumor microenvironments. Thus, our paper contributes an impressive amount of new and unrelated findings compared with the mentioned paper. Finally, we’d like to point out the distinct translational relevance of our work, fundamentally different from the previous paper, in that we showed the efficacy of the MAGL inhibitor JZL184 against self-renewal capacity of GSCs *in vitro* and *in vivo* (Fig. 7 of our manuscript). These results provide highly valuable translational impact of our findings with immediate possibility to benefit GBM patients in the clinical setting.
2. The story remains very convoluted, and the links between the steps, which are critical for the novelty of the paper, are still weak. For instance, the authors contend that the end product of ARS2

upregulation is signaling to cause M2 macrophage polarization, yet, the majority of the results in the paper take place in cell culture with no macrophages present, and the authors purport to show changes in stemness of glioma anyway. GSCs have been reported to cause M2 polarization (PMID: 25580734) so I'm unclear why this inclusion is necessary. Is ARS2 necessary for stemness, or for stem cells to induce M2 polarization, or just for survival of cells in general?

- Response: The first part of our paper is primarily focused on proving the functions of ARS2 and MAGL for glioma stemness. That's the reason why we conducted the experiments mostly with GSCs culture with no macrophages. Then, the latter part demonstrates cell to cell communication between GSCs and macrophages by modulating ARS2 and MAGL. To specifically answer this comment, we provide the evidence from animal experiments because the communication occurs not *in vitro*, but *in vivo*. However, to further support our results, here we provide direct evidence of our molecular mechanism in coculture system of GSCs (shCtrl, shARS2 or shMAGL) with bone marrow-derived macrophages (BMDMs). As shown in unpublished Fig. 1, expression of M2-like macrophage marker (CD206) was significantly decreased in BMDMs cocultured with ARS2- or MAGL-silenced GSCs. Therefore, our results present the direct evidence demonstrating ARS2/MAGL-mediated M2 polarization through cell to cell communication between GSCs and macrophages.

Unpublished Fig. 1. Expression of CD206 in BMDMs cocultured with GSC (X01) infected with a shARS2 #2, shMAGL-1 or shCtrl construct by flow cytometry. F4/80 was co-stained for macrophagic population verification.

3. The authors have not addressed the concern of artificial amplification of signal at each step. Specifically:

a. “Although ARS2 mRNA level was 1.5 times higher in glioma than normal tissue (Original Fig. 1a), we thought protein expression might not be exactly correlated. To address this issue, we quantified protein expression from cohort from National Cancer Center Korea (NCC) in original Fig. 1f and Supplementary Fig. 1. ARS2 protein was barely detectable in normal brains, but was widely and strongly expressed in patient tumor samples. These results support that our experimental overexpression of ARS2 is supposed to be within the clinically reproducible and reasonable alteration.” I’m confused by this point – the authors contend that the mRNA is not predictive of protein expression so figure 1A is either incorrect or misleading. Then why are the data still figure 1A? Also, the new data are slightly suspect – the authors suggest ARS2 expression in brain is undetectable, yet ARS2 is a common essential gene, with ARS2 disruption almost universally lethal (<https://depmap.org/portal/gene/SRRT?tab=overview>), including in CNS samples.

➤ Response: In generally, mRNA transcript of the genes is not always expressed in comparable level with protein expression. Much of the recent proteogenomic analysis demonstrated discrepancy between mRNA and protein (PMID: 30645970, 20023718, 23189060). Considering these limitations, we combined mRNA and protein expression data to show tendency of gene expression and survival rate. As reviewer pointed, overexpression system always has limitation of modulating physical relevance. To further support functional significance, we also added loss of function studies. Moreover, we conducted rescue experiment to prove functional importance of ARS2/MAGL axis (Fig. 4i, j).

b. “Concentration of PGE₂ treatment is also based on previous scientific references regulating gene expression on PGE₂ administration in macrophages at the level of 10 μM^{1,2}.” I don’t see the relevance of the references. PGE₂ should be added at concentrations that the glioma cells are capable of producing – levels which are never directly measured in the manuscript. What if the positive staining in figures 5a/b reflects nanomolar concentrations of PGE₂?

➤ Response: In the first decision letter, the reviewer commented, “Then the authors switch to directly adding high concentrations of PGE₂ to cells”. As we answered that “Concentration of PGE₂ treatment is also based on previous scientific references regulating gene expression on PGE₂ administration in macrophages at the level of 10 μM^{1, 2}. Based on the previous studies, we treated PGE₂ at three different levels up to the previously reported maximum concentration (0.1, 1, and 10 μM) and observed saturated response over 1 μM. Then, we conducted further experiments with 10 μM PGE₂, and we clearly observed full accumulation of β-catenin as shown in immunoblot staining (Original Fig. 5e).” To further validate PGE₂ working concentration in cell culture system, we measured PGE₂ concentration from GSC culture medium by ELISA method. As shown in unpublished Fig. 2, PGE₂

concentration in X01 control cells is around 7.27 μ M. Knockdown ARS2 or MAGL by shRNA reduced PGE₂ significantly. Our results demonstrate physiological concentration of PGE₂ secreted from GSC is enough to activate downstream signaling.

Unpublished Fig. 2. PGE₂ ELISA analysis in GSC (X01) infected with shARS2 #2, shMAGL-1 or shCtrl.

- The overreliance on immunofluorescence images is still unacceptable. The standard joke in science is that “representative” images means the best image you can find; without a more wholistic analysis the data are suspect. It is also unacceptable to force the reader to connect to a database with all of the higher visibility images – the manuscript should be a self-contained package that allows the reader to judge the work. Figures 6d-g are still problematic – even ignoring the doubts about the accuracy of the quantification raised by the previously incorrect results, the PGE₂ staining in figure 6d is definitely more red than figure 6f, yet the bar chart in 6g has a higher percent positive than 6e - showing the limitation of representative images.

➤ Response: As we mentioned that we are attaching our original images upon the request, we would like to present our raw data along with our cropped data. Both our original raw images and cropped images show highest quality.

The original images are showed in the below, and please see our revised Supplementary Figures page 14, 15, and 22-25.

Fig. 5a,b ICC staining (PGE₂)

Fig. 5a,b ICC staining (β -Catenin)

Fig. 5a,b. Original immunofluorescence images. Confocal microscopy images of PGE₂ and β -Catenin expression pattern of shCtrl, shMAGL-1, and shMAGL-3 of 528 and X01 cells. Scale bar, 50 μ m.

Fig. 6f ICC staining

Fig. 7d ICC staining

Fig. 6d,f and Fig. 7d. Original immunofluorescence images. Confocal microscopy images of PGE₂, CD86, CD206, ARG1, Nestin and GFAP of X01-shARS2 #2 (Fig. 6d), X01-shMAGL-1 (Fig. 6f), JZL184- or vehicle-treated X01 (Fig. 7d) in mouse brain tissues. Scale bar, 50 μm.

5. The authors response to problems with the LDAs are insufficient. LDAs are not designed to measure proliferation, but proliferation is a confounding factor in an LDA – anything that affects proliferation

will affect the results of an LDA. Figures 2c and 2f show that proliferation is greatly reduced by shARS2, so even if there is no effect on stemness, the LDA will show less sphere formation. This is where serial LDAs are critical – and a serial LDA involves taking the spheres formed by the initial LDA and replating them, not plating independent serial tissue culture passages for LDAs. Additionally, the time until reading should be consistent between experiments and 7 days is too short for adequate sphere formation.

➤ Response: Specific comment of the reviewer #2 in the first decision letter was “the authors would need to perform serial LDAs to show effects on sustained sphere formation. An analysis of non-stem glioma cells could also be potentially useful”. To address this, we performed serial LDAs until passage 3 and showed significant reduction of sphere forming ability in shARS2-infected GSCs, compared to shCtrl. To further verify specific function of ARS2 in GSC, we conducted proliferation assay with non-stem glioma cell, astrocytes infected with shARS2 #1 or #2. The proliferation assay demonstrated that proliferation of non-stem glioma cell, astrocytes was not significantly affected by shARS2 #1 or #2. These results demonstrate that ARS2 plays a critical role for cancer stemness and proliferation in GSCs but not in normal cells.

a. “As stemness grows stronger over time, we can easily expect that the stemness of the control X01 GSC in original fig. 4h harbors higher stemness.” This is an absolutely incorrect statement. Sphere formation increases with time, because of the effects of proliferation on the LDA, but those cells were always stem cells and the wells were false negatives until they were able to grow into a sphere. That’s why it’s critical to wait until saturation of sphere formation to get a readout of stem cell frequency independent of proliferation effects.

➤ Response: Thank you for your proper comments. We now provide new version of data measured at same timeline. For our revised Fig. 4h, we replaced our results from LDAs experiment with X01-shCtrl, X01-shMAGL-1, and X01-shMAGL-3, by which we counted well-formed spheres at the same time point (day 7). Please see revised Fig. 4d in our revised manuscript.

h, LDAs, performed in X01-shCtrl, X01-shMAGL-1, and X01-shMAGL-3 cells. The well-formed sphere was counted at day 16.

h, LDAs, performed in X01-shCtrl, X01-shMAGL-1, and X01-shMAGL-3 cells. The well-formed sphere was counted at day 7.

6. The rebuttal states that β -catenin is highly expressed in the nucleus rather than the cytosol, yet the staining in figure 5a/b is clearly cytoplasmic.
- Response: We appreciate reviewer's question. We also had similar concerns about location of β -catenin. To further address this question, we performed fractionation assay and 3D-image analysis. Our new data clearly demonstrate nuclear localization of β -catenin in GSCs.

528 shCtrl
 β -catenin
3D-Confocal
Images

X01 shCtrl
 β -catenin
3D-Confocal
Images

7. “Genes with statistical significance (p-value and q-value with less than 0.01) and fold-change (± 1.5) were chosen as significant DEGs.” All of the samples on the table except AL593851.1 and MGLL have a q-value above 0.01? Also, was ARS2 itself explicitly excluded, or does it not show up as significantly altered?
- Response: As we reported in the previous rebuttal letter, we have chosen the following DEGs based on three criteria: (1) statistical significance of p-value (< 0.01), (2) q-value (< 0.01), and (3) fold-change between ± 1.5 , by using SeqMonk software and LPEseq. Because MGLL met all the conditions, we regarded MGLL as a possible candidate. In consideration of q-value, as reviewer mentioned, only AL593851.1 and MGLL satisfied the significance. Reviewing p-value and valid fold-change, we detected 12 and 758 genes, which levels significantly altered by ARS2-knockdown in the analyses of SeqMonk and LPEseq, respectively. Furthermore, we verified ARS2-knockdown efficacy in the same shARS2 samples by RT-PCR and qPCR.

Reviewer #3 (Remarks to the Author):

The authors already did an effort to accommodate most of my concerns, but there are still a few important problems:

1) The authors changed peritoneal macrophages into bone marrow-derived macrophages for their *in vitro* experiments, which is already some improvement, but BMDM are still quite distinct from real TAM. As already asked before, it would be much more relevant and informative to sort real TAM (CD11b⁺ F4/80⁺) from ARS2 or MAGL KD tumors and treat those cells with PGE₂.

➤ **Response:** We thank the referee for these comments and we appreciate the importance of using real TAMs in addition to the bone marrow-derived macrophages (BMDMs) we used in the paper. As suggested, we presented data from CD11b⁺, F4/80⁺ TAMs that sorted from mouse tumor. FACS-sorted TAMs (CD11b⁺, F4/80⁺) from shCtrl or shARS2 tumors, treated with PGE₂, increased M2-like TAM marker CD206 expression, which supported our theory of ARS2/MAGL modulating M2-like TAMs via PGE₂ (new supplementary Fig. 6g). CD206⁺ TAM population was significantly decreased in shARS2 tumors compared to shCtrl tumors, but PGE₂ treatment has restored the CD206⁺ TAM population. Our results further support functional significance of ARS2/MAGL signaling of GSC in TAM education *in vivo*.

New supplementary Fig. 6g. Expression of CD206 in macrophages treated with PGE₂ or vehicle control for 24hrs by flow cytometry. CD11b and F4/80 was co-stained for macrophagic population verification.

- Page 11 line 7 – 14: To further demonstrate PGE₂-induced M2-like TAM polarization, we sorted CD11b and F4/80 double positive TAMs from mouse xenograft model, regarded as more relevant to real TAMs, extracted from ARS2-knockdown orthotopic xenograft mouse, and further confirmed the effect of PGE₂ on inducing M2-like polarization in the sorted TAMs. PGE₂ treatment on the FACS-sorted TAM from shARS2 and shMAGL tumors increased M2-like TAM marker CD206 expression, which supported our theory of ARS2/MAGL modulating M2-like TAMs via PGE₂ (Supplementary Fig. 6g).

2) The authors could use inhibitors that block PGE₂ production *in vivo*, to test the *in vivo* relevance of this pathway in their models.

- Response: As suggested by the reviewer, we included the new *in vivo* experiments in which mice were treated with PGE₂ synthesis inhibitor, celecoxib, in GSCs subcutaneous mouse xenograft model. As shown in new supplementary Fig. 7b-d, celecoxib significantly decreased tumor size and volume, and modulated M1- and M2-like TAMs, and stemness markers expression like as the effect of JZL184 on the paper.

New supplementary Fig. 7b-d. Celecoxib inhibits GSC self-renewal and tumorigenicity. b, Subcutaneous tumor size comparison analysis of mice implanted with X01 treated with Celecoxib 25mg/kg or vehicle (n=4, 3X10⁶ cells injected per mouse). c, Images of subcutaneous tumors model.

d, Representative IF images of PGE₂, CD86, CD206, ARG1, Nestin and GFAP in a celecoxib-treated subcutaneous mouse model. Sale bar, 50μM.

- Page 13 line 15 – 20: Furthermore, we blocked PGE₂ production by treating PGE₂ synthesis inhibitor, celecoxib in subcutaneous mouse xenograft model. As shown in supplementary Fig. 7b-d, celecoxib significantly decreased tumor size and volume, modulated M1- and M2-like TAMs, and reduced expression of stemness markers same as the effect of JZL184. These beneficial effects of JZL184 or celecoxib treatment indicate that pharmacological inhibition of MAGL or COX2 to block PGE₂ production suppresses the self-renewal and tumorigenic capacity of GSCs and promotes M1-like polarization of TAMs.
- 3) RNAseq to screen for macrophage genes was performed on cell preparations that contained less than 10% macrophages. This is unacceptable. How can the authors be sure that the regulated genes are intrinsic to macrophages? At the very least, genes whose expression were altered should be tested on purified TAM populations.
- Response: We highly appreciate the reviewer's comment. As the reviewer suggested, we potentiated our result by performing purification of macrophage with both CD11b⁺ and F4/80⁺ markers. Extracting samples from orthotopic mouse xenograft model, we conducted new RT-PCR on the purified TAM population (doubled sorted with CD11b⁺ and F4/80⁺). As emerged from our previous RNA-seq data which was sorted by only CD11b⁺, our new RT-PCT data validates once again the highest changes in the selected genes. Furthermore, JZL184 treatment decreased M2-like TAM genes expression, corresponding to the higher changes M2-TAM genes in our previous RNA-seq data.

New supplementary Fig. 8g,h. JZL184 downregulates M2-like TAMs markers expression. RT-PCR analysis of M2-like TAMs markers in CD11b⁺ (b) and CD11b⁺/F4/80⁺ (c) sorted macrophages treated JZL184 or vehicle. β -actin was used as a loading control.

- Page 14 line 5 – 8: We next examined the expression of the signature genes downregulated in the previously sorted CD11b⁺ macrophages, and also in the newly sorted CD11b and F4/80 double positive TAMs from subcutaneous mouse xenograft models. JZL184 treatment decreased M2-like TAM genes expression, significantly (Supplementary Fig. 8b,c).

Reviewers' Comments:

Reviewer #2:

Remarks to the Author:

The authors have made no progress in addressing the significant flaws with the manuscript. In fact, the only difference in the figures is that old figure 4h has been replaced with a worse version (seven days is not sufficient incubation time for neurosphere formation, as pointed out last review).

Reviewer #3:

Remarks to the Author:

The authors addressed my concerns

Reviewers' comments:

Reviewer #2 (Remarks to the Author):

Overall Comments: In the revised manuscript, the authors attempt to clarify their report, in which the transcription factor ARS2, previously shown to control SOX2 expression and neural stem cell identity and self-renewal, is linked to glioblastoma progression through transcriptional activation of MAGL causing production of prostaglandin E2 causing M2 tumor associated macrophage polarization driving a cancer stem cell phenotype. There were significant problems precluding publication of the initial submission, and the authors have made insufficient progress in addressing these flaws.

Specific Comments:

1. During the last revision period, a paper entitled “Ars2 promotes cell proliferation and tumorigenicity in glioblastoma through regulating miR-6798-3p” (PMID: 30349053) was published.
 - **Response:** First, we have noted that the paper mentioned by the referee (PMID: 30349053) was published in the journal *Scientific Report* on Oct 22, 2018 in the middle of our first-round review process. In first round reviewer’s comment, there were no comments related to this paper, so we thought possibly it’s because the contents of the published manuscript are fundamentally different from the works of our study. The main point of the published paper is elucidating role of ARS2 on target miRNA-793-3p exclusively in differentiated glioblastoma cell lines, which is already well-known function of ARS2. In contrast, we identify the ARS2-associated signaling mechanisms to drive self-renewal and tumorigenicity of GSCs, concurrently with M2-like TAM polarization. Also, the novel mechanisms in our study engage an original role as a transcription factor of ARS2 in GSCs and a number of previously unknown effectors communicating tumor microenvironments. Thus, our paper contributes an impressive amount of new and unrelated findings compared with the mentioned paper. Finally, we’d like to point out the distinct translational relevance of our work, fundamentally different from the previous paper, in that we showed the efficacy of the MAGL inhibitor JZL184 against self-renewal capacity of GSCs *in vitro* and *in vivo* (Fig. 7 of our manuscript). These results provide highly valuable translational impact of our findings with immediate possibility to benefit GBM patients in the clinical setting.
2. The story remains very convoluted, and the links between the steps, which are critical for the novelty of the paper, are still weak. For instance, the authors contend that the end product of ARS2

upregulation is signaling to cause M2 macrophage polarization, yet, the majority of the results in the paper take place in cell culture with no macrophages present, and the authors purport to show changes in stemness of glioma anyway. GSCs have been reported to cause M2 polarization (PMID: 25580734) so I'm unclear why this inclusion is necessary. Is ARS2 necessary for stemness, or for stem cells to induce M2 polarization, or just for survival of cells in general?

- Response: The first part of our paper is primarily focused on proving the functions of ARS2 and MAGL for glioma stemness. That's the reason why we conducted the experiments mostly with GSCs culture with no macrophages. Then, the latter part demonstrates cell to cell communication between GSCs and macrophages by modulating ARS2 and MAGL. To specifically answer this comment, we provide the evidence from animal experiments because the communication occurs not *in vitro*, but *in vivo*. However, to further support our results, here we provide direct evidence of our molecular mechanism in coculture system of GSCs (shCtrl, shARS2 or shMAGL) with bone marrow-derived macrophages (BMDMs). As shown in unpublished Fig. 1, expression of M2-like macrophage marker (CD206) was significantly decreased in BMDMs cocultured with ARS2- or MAGL-silenced GSCs. Therefore, our results present the direct evidence demonstrating ARS2/MAGL-mediated M2 polarization through cell to cell communication between GSCs and macrophages.

Unpublished Fig. 1. Expression of CD206 in BMDMs cocultured with GSC (X01) infected with a shARS2 #2, shMAGL-1 or shCtrl construct by flow cytometry. F4/80 was co-stained for macrophagic population verification.

3. The authors have not addressed the concern of artificial amplification of signal at each step. Specifically:

a. “Although ARS2 mRNA level was 1.5 times higher in glioma than normal tissue (Original Fig. 1a), we thought protein expression might not be exactly correlated. To address this issue, we quantified protein expression from cohort from National Cancer Center Korea (NCC) in original Fig. 1f and Supplementary Fig. 1. ARS2 protein was barely detectable in normal brains, but was widely and strongly expressed in patient tumor samples. These results support that our experimental overexpression of ARS2 is supposed to be within the clinically reproducible and reasonable alteration.” I’m confused by this point – the authors contend that the mRNA is not predictive of protein expression so figure 1A is either incorrect or misleading. Then why are the data still figure 1A? Also, the new data are slightly suspect – the authors suggest ARS2 expression in brain is undetectable, yet ARS2 is a common essential gene, with ARS2 disruption almost universally lethal (<https://depmap.org/portal/gene/SRRT?tab=overview>), including in CNS samples.

➤ Response: In generally, mRNA transcript of the genes is not always expressed in comparable level with protein expression. Much of the recent proteogenomic analysis demonstrated discrepancy between mRNA and protein (PMID: 30645970, 20023718, 23189060). Considering these limitations, we combined mRNA and protein expression data to show tendency of gene expression and survival rate. As reviewer pointed, overexpression system always has limitation of modulating physical relevance. To further support functional significance, we also added loss of function studies. Moreover, we conducted rescue experiment to prove functional importance of ARS2/MAGL axis (Fig. 4i, j).

b. “Concentration of PGE₂ treatment is also based on previous scientific references regulating gene expression on PGE₂ administration in macrophages at the level of 10 μM^{1,2}.” I don’t see the relevance of the references. PGE₂ should be added at concentrations that the glioma cells are capable of producing – levels which are never directly measured in the manuscript. What if the positive staining in figures 5a/b reflects nanomolar concentrations of PGE₂?

➤ Response: In the first decision letter, the reviewer commented, “Then the authors switch to directly adding high concentrations of PGE₂ to cells”. As we answered that “Concentration of PGE₂ treatment is also based on previous scientific references regulating gene expression on PGE₂ administration in macrophages at the level of 10 μM^{1, 2}. Based on the previous studies, we treated PGE₂ at three different levels up to the previously reported maximum concentration (0.1, 1, and 10 μM) and observed saturated response over 1 μM. Then, we conducted further experiments with 10 μM PGE₂, and we clearly observed full accumulation of β-catenin as shown in immunoblot staining (Original Fig. 5e).” To further validate PGE₂ working concentration in cell culture system, we measured PGE₂ concentration from GSC culture medium by ELISA method. As shown in unpublished Fig. 2, PGE₂

concentration in X01 control cells is around 7.27 μ M. Knockdown ARS2 or MAGL by shRNA reduced PGE₂ significantly. Our results demonstrate physiological concentration of PGE₂ secreted from GSC is enough to activate downstream signaling.

Unpublished Fig. 2. PGE₂ ELISA analysis in GSC (X01) infected with shARS2 #2, shMAGL-1 or shCtrl.

4. The overreliance on immunofluorescence images is still unacceptable. The standard joke in science is that “representative” images means the best image you can find; without a more wholistic analysis the data are suspect. It is also unacceptable to force the reader to connect to a database with all of the higher visibility images – the manuscript should be a self-contained package that allows the reader to judge the work. Figures 6d-g are still problematic – even ignoring the doubts about the accuracy of the quantification raised by the previously incorrect results, the PGE₂ staining in figure 6d is definitely more red than figure 6f, yet the bar chart in 6g has a higher percent positive than 6e - showing the limitation of representative images.
- Response: As we mentioned that we are attaching our original images upon the request, we would like to present our raw data along with our cropped data. Both our original raw images and cropped images show highest quality.

The original images are showed in the below, and please see our revised Supplementary Figures page 14, 15, and 22-25.

Fig. 5a,b ICC staining (PGE₂)

Fig. 5a,b ICC staining (β -Catenin)

Fig. 5a,b. Original immunofluorescence images. Confocal microscopy images of PGE₂ and β-Catenin expression pattern of shCtrl, shMAGL-1, and shMAGL-3 of 528 and X01 cells. Scale bar, 50 μm.

Fig. 6d ICC staining

Fig. 6f ICC staining

Fig. 7d ICC staining

Fig. 6d,f and Fig. 7d. Original immunofluorescence images. Confocal microscopy images of PGE₂, CD86, CD206, ARG1, Nestin and GFAP of X01-shARS2 #2 (Fig. 6d), X01-shMAGL-1 (Fig. 6f), JZL184- or vehicle-treated X01 (Fig. 7d) in mouse brain tissues. Scale bar, 50 μm.

5. The authors response to problems with the LDAs are insufficient. LDAs are not designed to measure proliferation, but proliferation is a confounding factor in an LDA – anything that affects proliferation

will affect the results of an LDA. Figures 2c and 2f show that proliferation is greatly reduced by shARS2, so even if there is no effect on stemness, the LDA will show less sphere formation. This is where serial LDAs are critical – and a serial LDA involves taking the spheres formed by the initial LDA and replating them, not plating independent serial tissue culture passages for LDAs. Additionally, the time until reading should be consistent between experiments and 7 days is too short for adequate sphere formation.

➤ Response: Specific comment of the reviewer #2 in the first decision letter was “the authors would need to perform serial LDAs to show effects on sustained sphere formation. An analysis of non-stem glioma cells could also be potentially useful”. To address this, we performed serial LDAs until passage 3 and showed significant reduction of sphere forming ability in shARS2-infected GSCs, compared to shCtrl. To further verify specific function of ARS2 in GSC, we conducted proliferation assay with non-stem glioma cell, astrocytes infected with shARS2 #1 or #2. The proliferation assay demonstrated that proliferation of non-stem glioma cell, astrocytes was not significantly affected by shARS2 #1 or #2. These results demonstrate that ARS2 plays a critical role for cancer stemness and proliferation in GSCs but not in normal cells.

a. “As stemness grows stronger over time, we can easily expect that the stemness of the control X01 GSC in original fig. 4h harbors higher stemness.” This is an absolutely incorrect statement. Sphere formation increases with time, because of the effects of proliferation on the LDA, but those cells were always stem cells and the wells were false negatives until they were able to grow into a sphere. That’s why it’s critical to wait until saturation of sphere formation to get a readout of stem cell frequency independent of proliferation effects.

➤ Response: Thank you for your proper comments. We now provide new version of data measured at same timeline. For our revised Fig. 4h, we replaced our results from LDAs experiment with X01-shCtrl, X01-shMAGL-1, and X01-shMAGL-3, by which we counted well-formed spheres at the same time point (day 7). Please see revised Fig. 4d in our revised manuscript.

h, LDAs, performed in X01-shCtrl, X01-shMAGL-1, and X01-shMAGL-3 cells. The well-formed sphere was counted at day 16.

h, LDAs, performed in X01-shCtrl, X01-shMAGL-1, and X01-shMAGL-3 cells. The well-formed sphere was counted at day 7.

6. The rebuttal states that β -catenin is highly expressed in the nucleus rather than the cytosol, yet the staining in figure 5a/b is clearly cytoplasmic.
- Response: We appreciate reviewer's question. We also had similar concerns about location of β -catenin. To further address this question, we performed fractionation assay and 3D-image analysis. Our new data clearly demonstrate nuclear localization of β -catenin in GSCs.

528 shCtrl
 β -catenin
 3D-Confocal
 Images

X01 shCtrl
 β -catenin
3D-Confocal
Images

7. “Genes with statistical significance (p-value and q-value with less than 0.01) and fold-change (± 1.5) were chosen as significant DEGs.” All of the samples on the table except AL593851.1 and MGLL have a q-value above 0.01? Also, was ARS2 itself explicitly excluded, or does it not show up as significantly altered?
 - Response: As we reported in the previous rebuttal letter, we have chosen the following DEGs based on three criteria: (1) statistical significance of p-value (< 0.01), (2) q-value (< 0.01), and (3) fold-change between ± 1.5 , by using SeqMonk software and LPEseq. Because MGLL met all the conditions, we regarded MGLL as a possible candidate. In consideration of q-value, as reviewer mentioned, only AL593851.1 and MGLL satisfied the significance. Reviewing p-value and valid fold-change, we detected 12 and 758 genes, which levels significantly altered by ARS2-knockdown in the analyses of SeqMonk and LPEseq, respectively. Furthermore, we verified ARS2-knockdown efficacy in the same shARS2 samples by RT-PCR and qPCR.

Reviewer #3 (Remarks to the Author):

The authors already did an effort to accommodate most of my concerns, but there are still a few important problems:

1) The authors changed peritoneal macrophages into bone marrow-derived macrophages for their *in vitro* experiments, which is already some improvement, but BMDM are still quite distinct from real TAM. As already asked before, it would be much more relevant and informative to sort real TAM (CD11b⁺ F4/80⁺) from ARS2 or MAGL KD tumors and treat those cells with PGE₂.

➤ **Response:** We thank the referee for these comments and we appreciate the importance of using real TAMs in addition to the bone marrow-derived macrophages (BMDMs) we used in the paper. As suggested, we presented data from CD11b⁺, F4/80⁺ TAMs that sorted from mouse tumor. FACS-sorted TAMs (CD11b⁺, F4/80⁺) from shCtrl or shARS2 tumors, treated with PGE₂, increased M2-like TAM marker CD206 expression, which supported our theory of ARS2/MAGL modulating M2-like TAMs via PGE₂ (new supplementary Fig. 6g). CD206⁺ TAM population was significantly decreased in shARS2 tumors compared to shCtrl tumors, but PGE₂ treatment has restored the CD206⁺ TAM population. Our results further support functional significance of ARS2/MAGL signaling of GSC in TAM education *in vivo*.

New supplementary Fig. 6g. Expression of CD206 in macrophages treated with PGE₂ or vehicle control for 24hrs by flow cytometry. CD11b and F4/80 was co-stained for macrophagic population verification.

- Page 11 line 7 – 14: To further demonstrate PGE₂-induced M2-like TAM polarization, we sorted CD11b and F4/80 double positive TAMs from mouse xenograft model, regarded as more relevant to real TAMs, extracted from ARS2-knockdown orthotopic xenograft mouse, and further confirmed the effect of PGE₂ on inducing M2-like polarization in the sorted TAMs. PGE₂ treatment on the FACS-sorted TAM from shARS2 and shMAGL tumors increased M2-like TAM marker CD206 expression, which supported our theory of ARS2/MAGL modulating M2-like TAMs via PGE₂ (Supplementary Fig. 6g).

2) The authors could use inhibitors that block PGE₂ production *in vivo*, to test the *in vivo* relevance of this pathway in their models.

- Response: As suggested by the reviewer, we included the new *in vivo* experiments in which mice were treated with PGE₂ synthesis inhibitor, celecoxib, in GSCs subcutaneous mouse xenograft model. As shown in new supplementary Fig. 7b-d, celecoxib significantly decreased tumor size and volume, and modulated M1- and M2-like TAMs, and stemness markers expression like as the effect of JZL184 on the paper.

New supplementary Fig. 7b-d. Celecoxib inhibits GSC self-renewal and tumorigenicity. b, Subcutaneous tumor size comparison analysis of mice implanted with X01 treated with Celecoxib 25mg/kg or vehicle (n=4, 3X10⁶ cells injected per mouse). c, Images of subcutaneous tumors model.

d, Representative IF images of PGE₂, CD86, CD206, ARG1, Nestin and GFAP in a celecoxib-treated subcutaneous mouse model. Sale bar, 50μM.

- Page 13 line 15 – 20: Furthermore, we blocked PGE₂ production by treating PGE₂ synthesis inhibitor, celecoxib in subcutaneous mouse xenograft model. As shown in supplementary Fig. 7b-d, celecoxib significantly decreased tumor size and volume, modulated M1- and M2-like TAMs, and reduced expression of stemness markers same as the effect of JZL184. These beneficial effects of JZL184 or celecoxib treatment indicate that pharmacological inhibition of MAGL or COX2 to block PGE₂ production suppresses the self-renewal and tumorigenic capacity of GSCs and promotes M1-like polarization of TAMs.

3) RNAseq to screen for macrophage genes was performed on cell preparations that contained less than 10% macrophages. This is unacceptable. How can the authors be sure that the regulated genes are intrinsic to macrophages? At the very least, genes whose expression were altered should be tested on purified TAM populations.

- Response: We highly appreciate the reviewer's comment. As the reviewer suggested, we potentiated our result by performing purification of macrophage with both CD11b⁺ and F4/80⁺ markers. Extracting samples from orthotopic mouse xenograft model, we conducted new RT-PCR on the purified TAM population (doubled sorted with CD11b⁺ and F4/80⁺). As emerged from our previous RNA-seq data which was sorted by only CD11b⁺, our new RT-PCT data validates once again the highest changes in the selected genes. Furthermore, JZL184 treatment decreased M2-like TAM genes expression, corresponding to the higher changes M2-TAM genes in our previous RNA-seq data.

New supplementary Fig. 8g,h. JZL184 downregulates M2-like TAMs markers expression. RT-PCR analysis of M2-like TAMs markers in CD11b⁺ (b) and CD11b⁺/F4/80⁺ (c) sorted macrophages treated JZL184 or vehicle. β -actin was used as a loading control.

- Page 14 line 5 – 8: We next examined the expression of the signature genes downregulated in the previously sorted CD11b⁺ macrophages, and also in the newly sorted CD11b and F4/80 double positive TAMs from subcutaneous mouse xenograft models. JZL184 treatment decreased M2-like TAM genes expression, significantly (Supplementary Fig. 8b,c).

Reviewers' Comments:

Reviewer #4:

Remarks to the Author:

Glioblastomas are known to have a myeloid cell enriched tumour microenvironment. Understanding how this is shaped by tumour signalling pathways is of great interest - particularly given the current failures of immunotherapy to have a major impact on this disease. Yin et al., explore the role played by glioblastoma stem cells in altering the phenotype of tumour-associated macrophages.

The authors focus on ARS2, a zinc finger transcription factor and suggest this has a major role in polarisation of TAMs - i.e. making them pro-tumorigenic. ARS2 is a transcription factor and this group suggests MGLL is a key target, which activates beta catenin via glioblastoma stem cell activation of prostaglandin.

These findings are potentially interesting, as Ars2 was identified in a Nature paper several years ago as a potential key regulator of Sox2 in mouse neural stem cells.

The authors perform knockdown of ARS2 in patient GSCs and identify MGLL from RNA-seq as a potential target gene. They show ARS2 is bound at the MGLL by Chip-seq.

A major concern of Reviewer 2 is: "Is ARS2 necessary for stemness, or for stem cells to induce M2 polarization, or just for survival of cells in general?" I agree that this is important to resolve. Is the self-renewal role critical and immune impact secondary, or both are critical?

Can this not be tested easily by transplanting into NSG mice, which are fully immunocompromised and comparing survival rates to the Nude mice transplants. i.e. what is the contribution of the cell autonomous self-renewal effects of ARS2 loss, versus the immune modulation role. They could easily dissect this.

I also agree that nuclear beta catenin staining and strength of that conclusion (which is difficult to infer from ICC) should be toned down or biochemical evidence presented.

Overall these are interesting insights, with a significant amount of experimental data, that identifies a role for ARS2 and influence on both beta catenin and TAMs via prostaglandin is both interesting tumour biology and has a potential therapeutic relevance.

minor points:

Macrophage signatures are already well described in the mesenchymal subtype of GBM. This is the subtype in which immune cells are increased. So this is not unexpected they see this.

Minor point - in the abstract it is not clear whether beta catenin is activated within the GSCs or TAMs or both.

REVIEWERS' COMMENTS:

Reviewer #4 (Remarks to the Author):

Glioblastomas are known to have a myeloid cell enriched tumour microenvironment. Understanding how this is shaped by tumour signalling pathways is of great interest - particularly given the current failures of immunotherapy to have a major impact on this disease. Yin et al., explore the role played by glioblastoma stem cells in altering the phenotype of tumour-associated macrophages.

The authors focus on ARS2, a zinc finger transcription factor and suggest this has a major role in polarisation of TAMs - i.e. making them pro-tumorigenic. ARS2 is a transcription factor and this group suggests MGLL is a key target, which activates beta catenin via glioblastoma stem cell activation of prostaglandin.

These findings are potentially interesting, as Ars2 was identified in a Nature paper several years ago as a potential key regulator of Sox2 in mouse neural stem cells.

The authors perform knockdown of ARS2 in patient GSCs and identify MGLL from RNA-seq as a potential target gene. They show ARS2 is bound at the MGLL by Chip-seq.

A major concern of Reviewer 2 is: "Is ARS2 necessary for stemness, or for stem cells to induce M2 polarization, or just for survival of cells in general?" I agree that this is important to resolve.

Can this not be tested easily by transplanting into NSG mice, which are fully immunocompromised and comparing survival rates to the Nude mice transplants. i.e. what is the contribution of the cell autonomous self-renewal effects of ARS2 loss, versus the immune modulation role. They could easily dissect this.

☞ We thank the reviewer for this comment. To address the future work required to clarify the role of ARS2 in the cross-talk between GSCs and macrophages, which can be directly observed in the highly specialized macrophages-immunodeficient mice as suggested by the reviewer, we have amended the argument to highlight this point. Specifically, on page 12 of the revised manuscript, we have updated the following sentences in the result section.

☐ Page 13 line 19 – line 22

“While our *in vitro* and *in vivo* data strongly supported the ability of ARS2 to concurrently drive glioma stemness and polarization into M2 macrophages, the complete elucidation of the details of ARS2 driven GSC-macrophages cross-talk will recessively require the use of highly specialized macrophages depleted immunodeficient mice.”

I also agree that nuclear beta catenin staining and strength of that conclusion (which is difficult to infer from ICC) should be toned down or biochemical evidence presented.

☞ We thank the reviewer for this comment, for we completely agree on the clarifying issue. We would be tone down in our manuscript, as indicated below.

□ Page 10 line 21 – line 23

Since expression of β -catenin is closely related with its role in transcriptional activation, we determined whether ARS2 or MAGL affects β -catenin expression and detected decreased nuclear β -catenin.

Overall these are interesting insights, with a significant amount of experimental data, that identifies a role for ARS2 and influence on both beta catenin and TAMs via prostaglandin is both interesting tumour biology and has a potential therapeutic relevance.

minor points:

Macrophage signatures are already well described in the mesenchymal subtype of GBM. This is the subtype in which immune cells are increased. So this is not unexpected they see this.

Minor point - in the abstract it is not clear whether beta catenin is activated within the GSCs or TAMs or both.

☞ We appreciate your comment and revised the manuscript as following.

□ Abstract

The interplay between glioblastoma stem cells (GSCs) and tumor-associated macrophages (TAMs) promotes progression of glioblastoma multiforme (GBM). However, the detailed molecular mechanisms underlying the relationship between these two cell types remain unclear. Here, we demonstrate that ARS2 (arsenite-resistance protein 2), a zinc finger protein that is essential for early mammalian development, plays critical roles in GSC maintenance and M2-like TAM polarization. ARS2 directly activates its novel transcriptional target *MGLL*, encoding monoacylglycerol lipase (MAGL), to regulate the self-renewal and tumorigenicity of GSCs through production of prostaglandin E₂ (PGE₂), which stimulates β -catenin activation of GSC and M2-like TAM polarization. We identify M2-like signature downregulated by which MAGL-specific inhibitor, JZL184, increased survival rate significantly in the mouse xenograft model by blocking PGE₂ production. Taken together, our results suggest that blocking the interplay between GSCs and TAMs by targeting ARS2/MAGL signaling offers a potentially novel therapeutic option for GBM patients.